# A Large-scale Universal Evaluation Benchmark For Face Forgery Detection

## Abstract

With the rapid development of AI-generated content (AIGC) technology, the production of realistic fake facial images and videos that deceive human visual perception has become possible. Consequently, various face forgery detection techniques have been proposed to identify such fake facial content. However, evaluating the effectiveness and generalizability of these detection techniques remains a significant challenge. To address this, we have constructed a large-scale evaluation benchmark called DeepFaceGen, aimed at quantitatively assessing the effectiveness of face forgery detection and facilitating the iterative development of forgery detection technology. DeepFaceGen consists of $776,990$ real face image/video samples and $773,812$ face forgery image/video samples, generated using $34$ mainstream face generation techniques. During the construction process, we carefully consider important factors such as content diversity, fairness across ethnicities, and availability of comprehensive labels, in order to ensure the versatility and convenience of DeepFaceGen. Subsequently, DeepFaceGen is employed in this study to evaluate and analyze the performance of $20$ mainstream face forgery detection techniques from various perspectives. Through extensive experimental analysis, we derive significant findings and propose potential directions for future research. The code and dataset for DeepFaceGen are available at https://anonymous.4open.science/r/DeepFaceGen-47D1.

## 1 Introduction

In recent years, AIGC technology has experienced rapid development, significantly enhancing its capabilities in abstract concept learning and content generation. This technology has initiated a global wave of artificial intelligence advancements, fundamentally transforming industries such as media, entertainment, e-commerce, and education.

However, AIGC is a double-edged sword that, while revolutionizing production manner, also introduces new security risks. Zhao et al. (2023) highlighted that malicious individuals can exploit AIGC to forge and tamper with data, making it increasingly difficult to verify the authenticity of generated facial images and videos. This tampering complicates the pursuit of truth, erodes trust in multimedia information, and poses significant security threats to society. As a result, criminal activities such as financial scams, internet rumors, and identity theft have become increasingly widespread.

To address the misuse of deepfake facial technology, numerous researchers from both industry and academia have proposed various techniques for detecting face deepfakes. These techniques heavily rely on publicly available face deepfake datasets. Thus, high-quality datasets are the cornerstone for developing effective deepfake detection techniques. Recently, several deepfake datasets (Table 1) have been created using deepfake techniques to assist researchers in training and evaluating their detection methods. However, most current deepfake datasets focus on relatively outdated task-oriented based face forgery techniques.

Recently, OpenAI released DALL·E and Sora, which capable of generating prompt-guided images and videos from textual descriptions, sparking a wave of prompt-guided generation. This technology surpasses the limitations of using existing images or videos for task-oriented edits, adopting a generative approach to creating fake content. In quick succession, numerous outstanding AIGC products have emerged, achieving unprecedented levels of generative technology. While enhancing

productivity and creative efficiency, these advancements also pose significant challenges for deepfake detection research.

Therefore, some researchers adopt the diffusion based generation technology to build the image dataset for AIGC detection. These datasets primarily consist of general images and do not provide precise "face" category data, which lack significant diversity and richness in terms of facial variations. In terms of video datasets, there is a notable lack of deepfake video datasets that incorporate prompt-guided based face forgery techniques, which are crucial for advancing face deepfake detection research. The absence of the evaluation dataset has led to a gap in face deepfake detection research, causing it to fall behind the rapid advancements in deepfake technology.

To address above challenge, this paper presents DeepFaceGen, a comprehensive and versatile evaluation benchmark specifically developed for face forgery detection. The main goal of DeepFaceGen is to facilitate the advancement of face forgery detection techniques. The benchmark encompasses a substantial dataset consisting of $463,583$ real images, $313,407$ real videos, $350,264$ forgery images, and $423,548$ forgery videos. The forgery samples are generated using $34$ prevalent image/video generation techniques. Leveraging DeepFaceGen, we conduct a comprehensive evaluation of existing face forgery techniques, examining their performance across various aspects such as forgery manner, generation framework, and generalization ability. Through extensive experimentation, we uncover noteworthy insights that are anticipated to provide valuable guidance for face forgery detection tasks.

## 2 RELATED WORKS

In this section, we provide a comprehensive overview of the existing deepfake datasets, presenting detailed information summaries in Table 1. The survey of both face forgery technology and face forgery detection technology can be found in *Appendix A*.

Early face forgery detection datasets generally suffer from a limited variety of forgery methods and are constrained in both quantity and quality. UADFV is the first dataset designed for face forgery detection. It only contains $49$ fake videos generated with the FakeApp (2019) application. The construction of APFDD, Celeb-DF, and DeeperForensics has significantly increased the scale of face forgery detection datasets. However, these datasets still only contain a single forgery method. To enrich the variety of forgery techniques in datasets, Korshunov & Marcel (2018) developed DeepfakeTIMIT using two face swapping techniques. Subsequently, Rossler et al. (2019) created FF++ using a total of four forgery methods: Deepfake, Face2face, Faceswap, and NeuralTextures. However, the size and diversity of FF++ are still insufficient, making it challenging to optimally train high-performance deep models with a large number of parameters. Zi et al. (2020) collected deepfake samples from the internet to create WildDeepfake, which includes facial motion sequences extracted from videos. After manually removing videos without corresponding real faces, the number of fake videos stands at $3,509$. Although the visual effects are closer to real-life scenarios, the limited data volume poses constraints on training high-performance deep models.

To address the issues of poor generation quality and coarse tampering traces in early face forgery detection datasets, DFDC, initially released as part of Facebook's eponymous competition, contains $5,250$ videos, which was later supplemented to reach $104,500$ fake videos generated using eight different methods to ensure dataset diversity. Following this, Kwon et al. (2021), Khalid et al. (2022), Zhou et al. (2021), and Narayan et al. (2023) addressed the limitations of existing datasets in terms of limited data diversity and content uniformity. They refined their datasets by focusing on factors such as racial diversity, multi-face scenes, and the granularity of labeling. In addition, He et al. (2021) developed ForgeryNet, the first face forgery detection dataset that includes both videos and images. They employed $15$ deepfake methods to generate $121,617$ fake videos and $1,457,861$ fake images. While these datasets significantly enhance both the quantity and quality of forgery methods, they remain limited to task-oriented techniques. This limitation makes them inadequate for detecting emerging AIGC-based forgery methods, which leverage prompt-guided generation, a more advanced and flexible approach to creating synthetic content.

The rapid development of prompt-guided generation based face forgery techniques, exemplified by diffusion, has led to the emergence of outstanding AIGC products such as Sora and DALL·E. These products have significantly impacted the field with their astonishing realism. The construction of deepfake datasets based on prompt-guided generation techniques has become increasingly urgent due

Table 1: Summary of existing deepfake datasets. * The authors of WildDeepfake note that the forged data was sourced from the internet, leaving the specific forgery methods unknown.

| Dataset Name | Content | Forged Data | | Generation Manner | | Racial Balance | Fine-grained Annotation | Forgery Approaches | Public Availability |
|---|---|---|---|---|---|---|---|---|---|
| | | Image | Video | Task-oriented | Prompt-guided | | | | |
| APFDD (Gandhi & Jain, 2020) | Face | 5,000 | - | ✓ | × | × | × | 1 | × |
| DeepArt (Wang et al., 2023a) | Art | 73,411 | - | × | ✓ | × | × | 5 | ✓ |
| IEEE VIP Cup (Cozzolino et al., 2023) | General | 7,000 | - | ✓ | ✓ | × | × | 14 | × |
| DE-FAKE (Sha et al., 2023) | General | 60,000 | - | × | ✓ | × | × | 4 | × |
| GenImage (Zhu et al., 2023) | General | 1,350,000 | - | ✓ | ✓ | × | × | 8 | ✓ |
| DiffusionForensics (Wang et al., 2023b) | General | 232,000 | - | × | ✓ | × | × | 10 | ✓ |
| DeepFakeFace (Song et al., 2023) | Face | 90,000 | - | ✓ | ✓ | × | × | 3 | ✓ |
| DiffusionDeepfake (Bhattacharyya et al., 2024) | Face | 112,627 | - | × | ✓ | × | × | 2 | ✓ |
| CiFAKE (Bird & Lotfi, 2024) | General | 60,000 | - | × | ✓ | × | × | 1 | ✓ |
| DF3 (Ju et al., 2024) | Face | 46,476 | - | ✓ | ✓ | × | × | 6 | ✓ |
| UADFV (Matern et al., 2018) | Face | - | 49 | ✓ | × | × | × | 1 | × |
| DeepfakeTIMT (Korshunov & Marcel, 2018) | Face | - | 320 | ✓ | × | × | × | 2 | ✓ |
| FF++ (Rossler et al., 2019) | Face | - | 4,000 | ✓ | × | × | ✓ | 4 | ✓ |
| Celeb-DF (Li et al., 2020b) | Face | - | 5,639 | ✓ | × | × | ✓ | 1 | ✓ |
| DeeperForensics (Jiang et al., 2020) | Face | - | 10,000 | ✓ | × | × | × | 1 | ✓ |
| WildDeepfake (Zi et al., 2020) | Face | - | 3,509 | ✓ | × | × | × | * | ✓ |
| DFDC (Dolhansky et al., 2020) | Face | - | 104,500 | ✓ | × | × | × | 8 | ✓ |
| KoDF (Kwon et al., 2021) | Face | - | 175,776 | ✓ | × | × | × | 6 | × |
| FFIW (Zhou et al., 2021) | Face | - | 10,000 | ✓ | × | × | ✓ | 3 | × |
| FakeAVCeleb (Khalid et al., 2022) | Face | - | 19,500 | ✓ | × | × | × | 3 | ✓ |
| DF-Platter (Narayan et al., 2023) | Face | - | 132,946 | ✓ | × | × | × | 3 | × |
| ForgeryNet (He et al., 2021) | Face | 1,457,861 | 121,617 | ✓ | × | × | × | 15 | ✓ |
| **DeepFaceGen (ours)** | **Face** | **350,264** | **423,548** | ✓ | ✓ | ✓ | ✓ | **34** | ✓ |

to the astonishing realism of these AIGC products. Through the continuous efforts of researchers, several high-quality datasets have emerged, such as DeepArt, DE-FAKE, DiffusionForensics, DiffusionDeepfake, and CiFAKE. However, a comprehensive dataset is crucial for both evaluating and advancing the development of deepfake detection models. These datasets only cover prompt-guided generation within the diffusion framework and lack a complete evaluation benchmark that integrates both prompt-guided and task-oriented forgery methods. Building on this, IEEE VIP Cup, Genimage, DeepFakeFace, and DF3 have established evaluation benchmarks that incorporate both prompt-guided and task-oriented forgery techniques. However, IEEE VIP Cup and Genimage are general-purpose datasets and do not provide "face" category forgery data. The introduction of DeepFakeFace and DF3 addresses this gap, but they still suffer from limitations, as DeepFakeFace includes only 3 forgery methods, while DF3 contains 6. Given the complexity and diversity of AIGC generation techniques, these limited forgery methods present significant constraints.

## 3 EVALUATION DATASET CONSTRUCTION

In this section, we aim to construct a robust and extensive benchmark for the detection of face forgery. To accomplish this, we carefully consider a range of critical factors including the manner of generation, generation framework, content diversity, ethnic fairness, and label richness throughout the benchmark development process. Following this, we provide a detailed introduction to the methodologies employed for collecting and generating forged samples. Additionally, we introduce the authentic data sources utilized by DeepFaceGen. Lastly, we present a comprehensive summary of the detailed data information encompassed within DeepFaceGen.

To enhance the diversity of DeepFaceGen, we augment its dataset by incorporating a selection of pre-existing forged face samples alongside newly generated ones using popular image and video generation techniques. These collected samples adhere to the principle of ethnic fairness. Specifically, from references Li et al. (2020b) and He et al. (2021), we choose samples created through task-oriented techniques such as face swapping, face reenactment, and face alteration. Detailed information about these collected samples can be found in *Appendix B*.

### 3.1 FORGED FACE SAMPLE GENERATION

For the novel AIGC techniques, we employ a set of 17 prevalent prompt-guided generation based face forgery techniques. Additionally, we incorporate 17 classical task-oriented based face forgery techniques, excluding the new generation methods. In the following section, we extensively elaborate on the generation processes for both categories of techniques.

**Prompt-guided Based Generation** techniques utilize text or image input to generate prompt-guided samples. The design of the prompt plays a crucial role in determining the quality of the generation

outcome. Hence, we primarily present the process of prompt construction, followed by the description of forgery methods.

- **Prompts Construction**. In the design of prompts, we strive to achieve both content diversity and fairness, which are accompanied by a strong emphasis on detailed prompt descriptions. For each prompt, we establish fundamental attributes, such as age, gender, and skin tone, while also providing comprehensive specifications regarding the person's background and physical features. The inclusion of these extensive textual attribute details further facilitates the evaluation of forgery detection performance at a fine-grained level. A total of 9 textual attributes are defined in the prompt construction process. By exhaustively generating prompts using all possible combinations of these textual attributes, we ensure the creation of a diverse and equitable set of forged data. For further elaboration on these prompts, please refer to *Appendix C*.

- **Text2Image** generation techniques involve three main categories: GAN, autoregressive, and diffusion frameworks. Some of these techniques have been developed into commercial products. In order to enhance the practicality and universality of DeepFaceGen, we have incorporated mature commercial products and popular open-source methods to generate the forgery samples. For GAN-based models, we have adopted the popular open-source DF-GAN (Tao et al., 2022) which employs adversarial training between the generator and discriminator to achieve impressive image generation capabilities. As for autoregressive based models, we have utilized OpenAI's commercial product DALL·E and DALL·E 3 (Open AI, 2023), which treats text tokens and image tokens as a unified data sequence and uses a Transformer for auto-regression. Given that existing high-quality generation techniques mostly rely on diffusion framework, we have incorporated specific models such as OpenAI's Midjourney (Midjourney, 2022), Baidu's Wenxin (Baidu, 2022), Stability.ai's series products {Stable Diffusion 1 (SD1), Stable Diffusion 2 (SD2), Stable Diffusion XL (SDXL)} (Stability.ai, 2023), and PromptHero's open-source version of Midjourney (Openjourney, OJ) PromptHero (2023).

- **Image2Image** generation involves utilizing an image as input to generate prompt-guided samples, typically employing diffusion frameworks. Therefore, we utilize Stable Diffusion XL Refiner (SDXLR), Stable Diffusion InstructPix2Pix (Pix2Pix), and Stable Diffusion ImageVariation (VD) (Stability.ai, 2023), all of which have achieved high rankings on Huggingface's download charts.

- **Text2Video** techniques involve using a text prompt as input to generate a complete video sample, also relying on diffusion frameworks. However, due to unavailability of certain mature commercial products' API, we have selected alternative products. Specifically, we have chosen MagicTime (Yuan et al., 2024), AnimateDiff-Lightning (AnimateDiff) (Lin & Yang, 2024), AnimateLCM (Wang et al., 2024), Hotshot (Mullan et al., 2023), and Zeroscope (Academy for Discovery, 2023).

**Task-oriented Based Generation** technique generates forged samples by modifying certain parts of input face images. Existing task-oriented techniques can be categorized into three types: face swapping, face reenactment, and face alteration.

- **Face Swapping** technique involves creating a manipulated face sample by exchanging the faces of two given image samples. In this study, we employ 8 commonly used face swapping methods, namely FaceShifter (Li et al., 2019), FSGAN (Nirkin et al., 2019), DeepFake (Faceswap, 2020), BlendFace (Shiohara et al., 2023), MMReplacement (He et al., 2021), DeepFakes-StarGAN-Stack (DSS), StarGAN-BlendFace-Stack (SBS), and SimSwap (Chen et al., 2020). Among these approaches, DSS and SBS are categorized as mixed face forgery methods, wherein the face alteration technique is initially applied before face swapping is performed.

- **Face Reenactment** technique involves transferring the facial movements and expressions from one person onto the face of another person. In this study, we utilize four specific approaches for face reenactment: Talking Head Video (Fried et al., 2019), ATVG-Net (Chen et al., 2019), FOMM (Siarohin et al., 2019a), and Motion-cos (Siarohin et al., 2020).

- **Face Alteration** technique involves creating forged images by making subtle modifications to facial attributes such as hair color, beard, and glasses. The face alteration approaches

utilized in this study include StyleGAN2 (Karras et al., 2019), MaskGAN (Lee et al., 2019), StarGAN2 (Choi et al., 2019), SC-FEGAN (Jo & Park, 2019), and DiscoFaceGAN (Deng et al., 2020).

## 3.2 AUTHENTIC FACE SAMPLE COLLECTION

In order to ensure content diversity and ethnic fairness in the authentic face samples used in Deep-FaceGen, we obtained real samples from reputable sources including Li et al. (2020b), He et al. (2021), Chen et al. (2023), and Zhao et al. (2019). The final collection consists of $463, 583$ images and $313, 407$ videos, encompassing diverse races, genders, ages, expressions, hairs, backgrounds, and so on. Please refer to *Appendix B* for more details.

## 3.3 DATASET SUMMARIZATION

The aforementioned generation and collection processes yield the initial dataset samples. To ensure both sample quality and racial balance, postprocess operations are implemented to filter these samples. The SkinToneClassifier (Pia & Ma, 2023) is utilized for racial balance, ensuring skin tone balance in the generation and collection of task-oriented based and Image2Image face forgery methods. For prompt-guided generation-based face forgery techniques (Text2Image and Text2Video), the combination and design of text prompts also take skin tone balance into consideration. Based on fine-grained annotation, we explore the difference in detection performance of the detectors in nine attributes and reach several constructive conclusions. Please refer to *Appendix H* for more details.

Additionally, we used YOLO (ultralytics, 2020) to score and filter generated fake images/videos, removing those that fell below a set threshold. Low-quality data was then manually discarded, resulting in a dataset of "realistic" samples capable of deceiving the human eye. Deepfacegen achieved an FID score of 28.85 (where lower values indicate higher realism), which is significantly better than the scores of ForgeryNet (36.94), DiffusionForensics (31.79), and FF++ (33.87). These measures effectively maintain the fairness and reliability of DeepFaceGen, resulting in the collection of $350, 264$ forged images and $423, 548$ forged videos. For a detailed breakdown of the sample numbers for different generation techniques, please refer to Figure 2 provided in *Appendix B*.

## 4 BENCHMARK EVALUATION AND ANALYSIS

In this section, we employ DeepFaceGen to evaluate 20 prevalent face forgery detection methods from various perspectives, such as generation approach type, generalization capability, and technique relevance. Subsequently, we analyze extensive experimental results and summarize key findings, elucidating the strengths and weaknesses of current face forgery detection techniques, as well as identifying potential directions for future research.

**Evaluation Settings.** Based on the distinction in modality between images and videos, we partition DeepFaceGen into two parts. The image and video datasets are divided into training, validation, and test subsets in a ratio approximately $7 : 1 : 2$. To ensure fairness in evaluation, each subset maintains a ratio of real to fake instances close to $1 : 1$. For image-level assessments, we employ Xception (Chollet, 2017), EfficientNet-B0 (Tan & Le, 2020), F3-Net (Qian et al., 2020b), RECCE (Cao et al., 2022b), DNADet (Yang et al., 2022), DIRE (Wang et al., 2023b),DRCT (Chen et al., 2024),UnivFD (Ojha et al., 2023),NPR (Tan et al., 2024a),and FreqNet (Tan et al., 2024b). For video-level evaluations, we select MesoNet (Afchar et al., 2018), EfficientNet-B0 (Tan & Le, 2020), Xception (Chollet, 2017), F3-Net (Qian et al., 2020b), CViT (Wodajo & Atnafu, 2021), SLADD (Chen et al., 2022), TALL (Xu et al., 2023),AltFreezing (Wang et al., 2023c),Exposing (Ba et al., 2024),and LSDA(Yan et al., 2024b), as they exhibit exceptional performance in forgery video detection. The experiments are conducted separately on Nvidia A40 GPU (48GB VRAM) and two machines, each featuring a GeForce RTX 4090 GPU (24GB VRAM). More evaluation details are given in the *Appendix D*.

## 4.1 EVALUATION OF MAINSTREAM FORGERY DETECTION TECHNIQUES

In this section, we initiate the training of all forgery detection models utilizing training samples obtained from DeepFaceGen. We subsequently present and analyze the experimental results com-

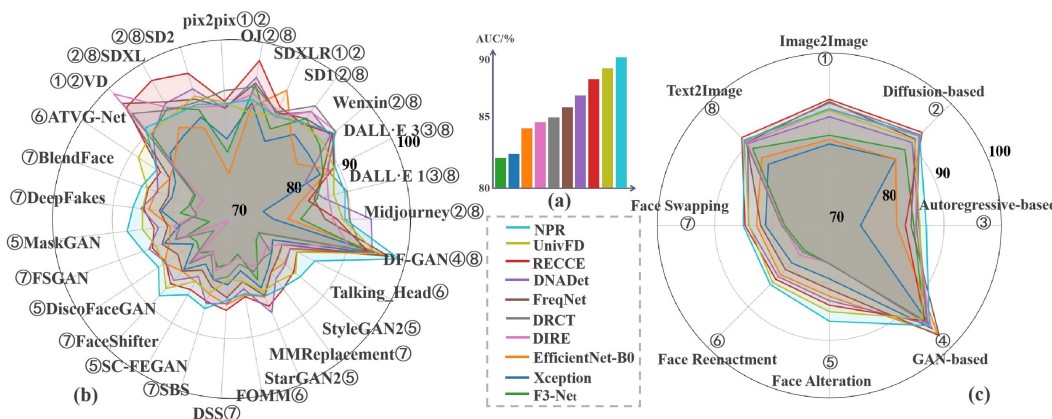

Figure 1: Image-level Performance comparison of different forgery detection techniques. (a) Average detection performance ranking. (b) Detection performance for different generation techniques. (c) Detection performance for different generation manners and frameworks (marked with ①-⑧).

prehensively, considering various aspects such as the sample modality, forgery technique, forgery technique type, and the framework employed by the forgery detection models.

### 4.1.1 IMAGE-LEVEL EVALUATION AND ANALYSIS

**Forgery Detection Technique Comparison.** Figure 1 (a) illustrates the average detection performance of various forgery detection techniques. As shown in the figure, NPR (Tan et al., 2024a), RECCE (Cao et al., 2022b), and UnivFD (Ojha et al., 2023) outperform the other methods, while Xception (Chollet, 2017), EfficientNet (Chollet, 2017), and F3-Net (Qian et al., 2020b) demonstrate poor performance. NPR captures and characterizes local pixel dependencies in images using up-sampling operators. By quantifying the dependencies between neighboring pixels, it constructs features that represent local pixel differences, which are not limited to specific forgery methods, achieving excellent results in deepfake detection. Similarly, UnivFD recognizes the importance of extracting fine-grained details and utilizes a pre-trained CLIP model to map images into feature representations for forgery identification. Additionally, RECCE employs a custom Encoder-Decoder structure with a multi-scale graph reasoning module to capture feature representations. These three methods leverage their respective architectures to extract detailed features related to forgery. In contrast, general-purpose classifiers like Xception (Chollet, 2017), F3-Net (Qian et al., 2020b), and EfficientNet-B0 (Tan & Le, 2020), which utilize convolutional encoder architectures, perform worse compared to specialized methods designed for face forgery detection. Thus, it can be concluded that *the detail extraction module plays a critical role in the detection of face image forgery (**Finding 1**).* Further details on the detail extraction module are provided in *Appendix E*.

**Generation Manner and Framework.** Based on Figure 1 (c), it is evident that task-oriented techniques (face swapping, face reenactment, face alteration) for image generation can produce more challenging identification samples compared to prompt-guided generation techniques (Text2Image and Image2Image). This can be attributed to *the relative ease of generating authentic images by modifying smaller localized areas rather than the entire image (**Finding 2**).* However, further research is required to enhance the performance of prompt-guided generation techniques. Regarding different generation frameworks, it is evident that autoregressive based techniques (DALL·E and DALL·E3 (Open AI, 2023)) achieve the highest quality of forgery, surpassing diffusion-based and GAN-based techniques. The newly proposed diffusion-based framework demonstrates the second-best average performance, indicating its potential for further development. Conversely, GAN-based generation techniques exhibit the poorest quality for forgery. Therefore, it can be concluded that *autoregressive-based and diffusion-based generation techniques are capable of producing more realistic forged face images than GAN-based generation techniques (**Finding 3**).*

**Input Modality.** Based on the results depicted in Figure 1 (b)(c), it is apparent that both Text2Image (Midjourney (Midjourney, 2022), OJ (PromptHero, 2023), SD1 (Stability.ai, 2023), SD2 Stability.ai

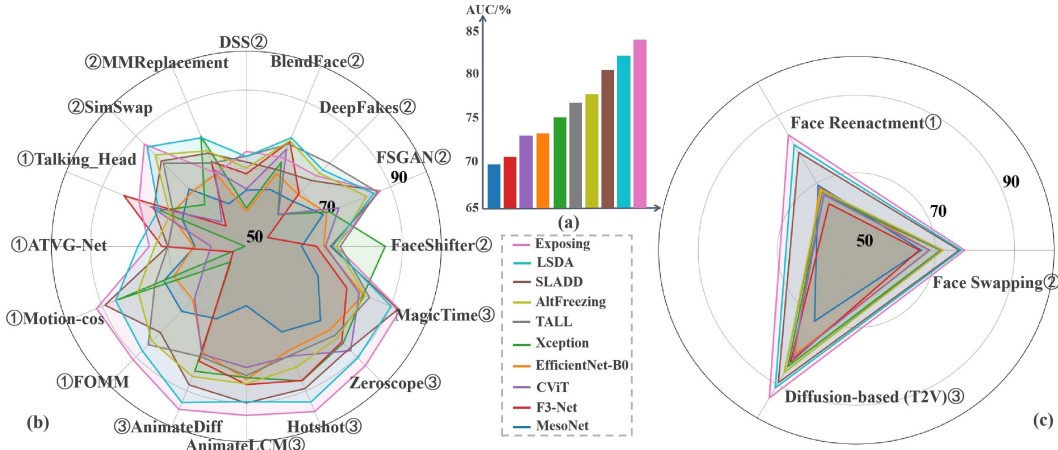

Figure 2: Video-level Performance comparison of different forgery detection techniques. (a) Average detection performance histogram. (b) Detection performance for different generation techniques. (c) Detection performance for different generation manners and frameworks (marked with ①-③).

(2023) and SDXL (Stability.ai, 2023)) and Image2Image techniques (SDXLR (Stability.ai, 2023), Pix2Pix (Stability.ai, 2023), and VD Stability.ai (2023)) that employ the same diffusion-based framework deliver comparable performance. Consequently, it can be deduced that *the choice of input modality has minimal influence on the quality of image generation (**Finding 4**)*.

### 4.1.2 VIDEO-LEVEL EVALUATION AND ANALYSIS

**Forgery Detection Technique Comparison.** Figure 2 (a) depicts the average detection performance of various video forgery detection techniques. It is evident that both Exposing (Ba et al., 2024), LSDA (Yan et al., 2024b) and SLADD (Chen et al., 2022) outperform the rest. Exposing adopts a two-step approach: extracting frame-level facial bounding boxes from raw videos and subsequently extracting multiple disentangled local features from different regions for forgery detection. LSDA learns a more generalizable decision feature by expanding the forgery space, constructing and simulating variations of forgery features within the latent space. This process helps the extraction of enriched, domain-specific features and facilitates smoother transitions between different forgery types, effectively bridging the domain gaps. SLADD employs adversarial self-supervised training to identify various forgery detail features, which contributes to its outstanding performance. In contrast, general-purpose classifiers such as EfficientNet-B0 (Tan & Le, 2020), Xception (Chollet, 2017), F3-Net (Qian et al., 2020b), and CViT (Wodajo & Atnafu, 2021) exhibit poor identification performance due to their lack of forgery detail information. Thus, we can conclude that the *extraction of detailed features also plays a critical role in detecting face video forgery (**Finding 5**)*.

**Generation Manner and Framework.** This study focuses on high-quality prompt-guided video generation techniques and predominantly adopts the diffusion-based framework. Methods (Saito et al., 2017; Clark et al., 2019; Yan et al., 2021) with poor visual video generation quality are not included in this investigation. Analysis of Figure 2 (b) reveals that prompt-guided generation techniques with diffusion framework demonstrate similar performance. Consequently, we can infer that *existing diffusion-based generation techniques possess a comparable ability to generate forged videos (**Finding 6**)*. Additionally, diffusion-based techniques exhibit lower performance compared to alternative methods, with face swapping yielding the best results. The potential explanation for this finding is that diffusion-based techniques, face reenactment, and face swapping alter the content of the full image, facial movements, and facial contour, respectively. Consequently, it can be inferred that *altering fewer aspects of content leads to the generation of more authentic videos (**Finding 7**)*.

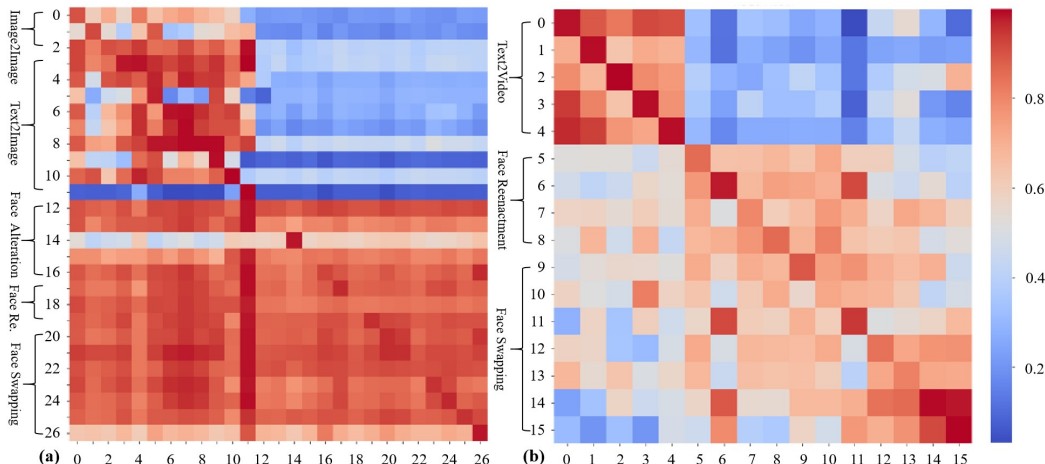

Figure 3: The cross-generalization ability verification matrices for image-level (a) and video-level (b) datasets. The training and testing samples, generated by various forgery techniques, are represented on the vertical and horizontal axes. The denotation for each number is provided in the *Appendix F*.

## 4.2 GENERALIZATION ABILITY EVALUATION TO DIFFERENT FORGERY TECHNIQUES

In this section, we verify the cross-generalization ability among sub-datasets created using various forgery techniques. The results for image-level and video-level datasets, obtained using the Xception model for forgery detection, are presented in Figure 3. Furthermore, additional cross-generalization verification experiments with another 18 forgery detection models can be found in *Appendix F* .

**Generalization Ability Across Different Forgery Techniques.** Figure 3 demonstrates that models trained on task-oriented forgery images/videos exhibit superior generalization capability than models trained on prompt-guided forgery images/videos. This difference can be attributed to several factors. Task-oriented forgery techniques concentrate on specific facial regions, such as eyes, mouth, and skin texture, which also serve as vital clues for detecting prompt-guided forgery images/videos. Conversely, prompt-guided forgery methods consider the entire image, incorporating elements like background, lighting, and environment, which introduce significant variability across different datasets. Consequently, the model's ability to generalize on task-oriented samples is diminished. Thus, we can conclude that *Face forgery detection methods trained on task-oriented samples generally demonstrate higher generalization capability compared to those trained on prompt-guided generation samples (**Finding 8**)*. Furthermore, from Figure 3(a), it is evident that the forgery detection technique trained and tested on samples generated by the prompt-guided DF-GAN (Tao et al., 2022) exhibits poor and good generalization ability, respectively. This finding further confirms **Finding 3** that prompt-guided generation using GAN-based techniques results in low image quality, making it easily detectable by forgery detection techniques.

**Internal Generalization Ability Analysis.** Figure 3 indicates that models trained on prompt-guided forgery samples (Image2Image, Text2Image, and Text2Video) possess a high degree of internal generalization ability. This can be attributed to the significant similarities shared among samples generated by prompt-guided generation techniques. Similarly, models trained on task-oriented forgery images and videos demonstrate high and moderate internal generalization ability, respectively. Moreover, models trained on face reenactment videos and face swapping forgery videos exhibit a moderate level of generalization ability to each other. The findings imply a trend in forgery detection methods, where generalized forgery features are learned from images, while more specific forgery features are acquired from videos. This disparity may be attributed to the presence of redundant features in videos compared to single images. Hence, we can conclude that *models trained on prompt-guided forgery images, task-oriented forgery images, and prompt-guided forgery videos display high internal generalization ability, whereas task-oriented forgery videos do not (**Finding 9**)*.

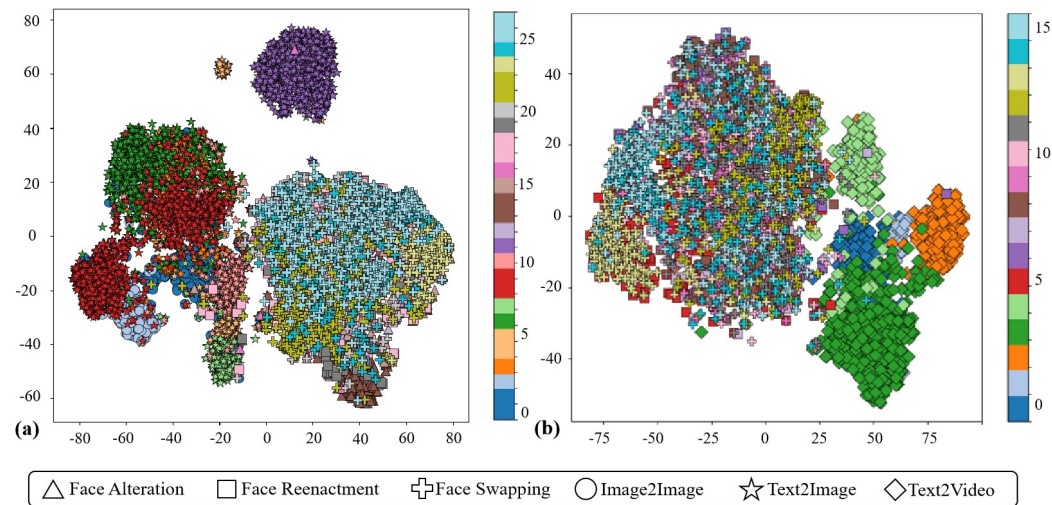

Figure 4: The forgery feature visualization for different forgey techniques on image-level (a) and video-level (b) datasets with t-SNE (van der Maaten & Hinton, 2008).

### 4.3 VISUALIZATION ANALYSIS OF FORGERY DETECTION FEATURES

In this section, we utilize the fully connected layer features of the forgery detection model ResNet50 (He et al., 2016) to visually evaluate the similarities among different forgery techniques. As illustrated in Figure 4, a clear distinction is observed in the feature space between prompt-guided forgery samples (Image2Image, Text2Image, and Text2Video) and task-oriented forgery samples (face alteration, face reenactment, and face swapping). This indicates that *the forgery features of prompt-guided forgery samples and task-oriented forgery samples are distinct (**Finding 10**). It further confirms the **Finding 8&9**. Further analysis is given in *Appendix G*.

## 5 CONCLUSION

In this study, we present DeepFaceGen, the first comprehensive deep face forgery dataset that encompasses both task-oriented and prompt-guided generation samples. This dataset addresses the existing gap in large-scale general face forgery datasets. DeepFaceGen contains an extensive collection of over $350,000$ images and $400,000$ videos. We provide a detailed description of the dataset construction process and evaluate the performance of 20 mainstream forgery detection techniques on samples forged using 34 different generation techniques. By analyzing the results of these extensive experiments, we draw important findings that present novel perspectives and directions for the development of face generation and forgery detection techniques. We anticipate that this benchmark will have a far-reaching positive impact on the emerging field of artificial intelligence.

**Challenge and Future Work.** Based on extensive experimentation and analysis, it is evident that current forgery detection techniques suffer from drawbacks, such as low identification accuracy, poor generalization ability, and a restricted range of forgery detection types. Moreover, the rapid development of face generation techniques has created a significant discrepancy, resulting in a lag in face forgery detection. In order to address this issue, the development of a self-evolving forgery detection framework is crucial to ensure that forgery detection techniques can keep up with the advancements in face generation techniques. Additionally, this paper presents a comprehensive evaluation benchmark comprising diverse content samples, various races, and fine-grained labeling. The design of objective and comprehensive quantification metrics, as well as the establishment of a complete pipeline, are crucial for future research. Further analysis regarding challenges and future directions can be found in the *Appendix I*.

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

APPENDIX

In the appendix, we provide survey of face forgery technology and face forgery detection technologies (A), comprehensive statistical analysis of the DeepFaceGen dataset (B), and the detailed descriptions of prompts construction (C). We also outline the evaluation setting details (D), details on the detail extraction module (E), details for generalization ability verification experiments of different methods (F), and fine-grained analysis of forgery detection feature (G). Additionally, we give fine-grained attribute statistic analysis for different forgery techniques (H), detailed challenge discussions and future directions (I) and potential negative social impacts (J).

# A SURVEY OF FACE FORGERY TECHNOLOGY AND FACE FORGERY DETECTION TECHNOLOGY

In this section, we present a comprehensive overview of both face forgery technologies and face forgery detection technologies. Regarding the former, we categorize face forgery methods into task-oriented and prompt-guided generation techniques based on their image/video generation approach. Subsequently, we discuss the forgery detection techniques designed specifically for these two types of forgery methods.

## A.1 TASK-ORIENTED BASED FACE FORGERY TECHNOLOGY

Task-oriented based face forgery involves modifying specific facial features, such as expressions and movements. Traditional facial Photoshop (PS) techniques, which involve manual image manipulation, also fall within this scope. However, traditional PS techniques often leave detectable traces that can be identified by the naked eye. Therefore, survey of task-oriented based face forgery focus on advanced deepfake methods including face swapping, face reenactment, and face alteration.

**Face Swapping.** Face swapping involves transferring the facial identity from a source image to a target image while preserving the expressions, movements, and background of the target image. Early face swapping techniques primarily relied on autoencoders. One such tool, Deepfake (Faceswap, 2020), popularized by Reddit users, trains the facial images of the source and target persons separately, allowing the decoder to accurately reproduce their faces. In face swapping, the encoder extracts the source person's facial features and inserts them into the target person's image using the decoder. Shaoanlu (2017) introduces FaceswapGAN, which employs a face swapping attention mechanism to enhance image realism. This method also addresses occlusion issues using segmentation masks. RSGAN (Natsume et al., 2018) is designed for face swapping using two autoencoders to represent the hair and face regions. It replaces the face's latent representation and reconstructs the image, effectively addressing issues such as mismatched face orientation and lighting. Nirkin et al. (2019) introduces FSGAN, which uses RNN-based methods to transfer expressions and movements from the target face to the source face. FSGAN demonstrates good generalization and requires fewer training samples. Li et al. (2019) introduces Faceshifter, a two-stage face-swapping method. It uses adaptive attention denormalization (AAD) for feature integration and employs a heuristic error acknowledgment refinement network (HEAR-Net) to address occlusion issues. Chen et al. (2020) introduces an identity injection module to eliminate identity constraints, and enhances the loss function with weak feature matching loss to improve face synthesis quality.

**Face Reenactment.** Face reenactment preserves the target image's facial identity while replicating expressions, facial orientation, and body movements from the source image.. Wang et al. (2020) introduces Imaginator, which uses a spatiotemporal feature fusion mechanism to decode continuous video from spatial features and motion. They employ two discriminators: one to evaluate the realism of facial appearances and the other to assess the realism of motions. Siarohin et al. (2019b) introduces Monkey-Net, which separates appearance and motion information in images, enabling motion-driven animation. Monkey-Net includes a motion transfer network, an unsupervised keypoint detector, and a motion prediction network. It predicts the visual flow map for each keypoint by distinguishing keypoints in target and source images, thereby generating forged images. Siarohin et al. (2019a) improves on Monkey-Net by introducing local affine transformations around keypoints, which better reproduce large pose variations. Pumarola et al. (2018) uses action unit annotations combined with unsupervised training and attention mechanisms to enhance model robustness. Tripathy et al. (2020) uses action units to represent facial expressions, processing the face and background separately to

improve image quality and reduce identity information leakage. CycleGAN (Zhu et al., 2017) is widely used in face reenactment due to its flexible training capabilities between source and target domains. Xu et al. (2017) proposes a full-image reenactment method based on CycleGAN, which uses various receptive field specifications and PatchGAN to enhance image quality. Bansal et al. (2018) uses CycleGAN for data-driven, unsupervised video retargeting, effectively transferring continuous information for expression-driven animation. Wu et al. (2018) introduces ReenactGAN, which extracts facial contours using an encoder and maps them via CycleGAN. A pix2pix generator then reconstructs the image. This method uses only feedforward neural networks, enabling real-time expression reenactment.

**Face Alteration.** Face alteration modifies specific attributes like hair color, gender, and glasses without altering facial identity. Most face alteration techniques use GAN structures. The StyleGAN series (Karras et al., 2018; 2019; 2021) are notable for editing facial features, while StarGAN (Choi et al., 2017) and StarGANV2 (Choi et al., 2019) enable transformations across multiple image domains, offering better scalability. Another notable method is GANnotation (Sanchez & Valstar, 2018), which contains a triple continuity loss function for GAN-based face alteration and a direct facial expression alteration synthesis method. Kim et al. (2021) introduces a CAM consistency loss function based on CycleGAN's cycle consistency loss function, which helps retain feature-independent positional information and can be applied to models like StarGAN. To address scalability and diversity issues in face alteration, Li et al. (2021) introduces hierarchical style disentanglement(HiSD), a hierarchical model that represents facial features as labels and attributes. Using an unsupervised approach, HiSD decouples these features, allowing for more precise modifications of target attributes.

A.2 PROMPT-GUIDED GENERATION BASED FACE FORGERY TECHNOLOGY

Based on the differences in network architecture, prompt-guided generation face forgery techniques can be categorized into gan-based models, autoregressive-based models, and diffusion-based models.

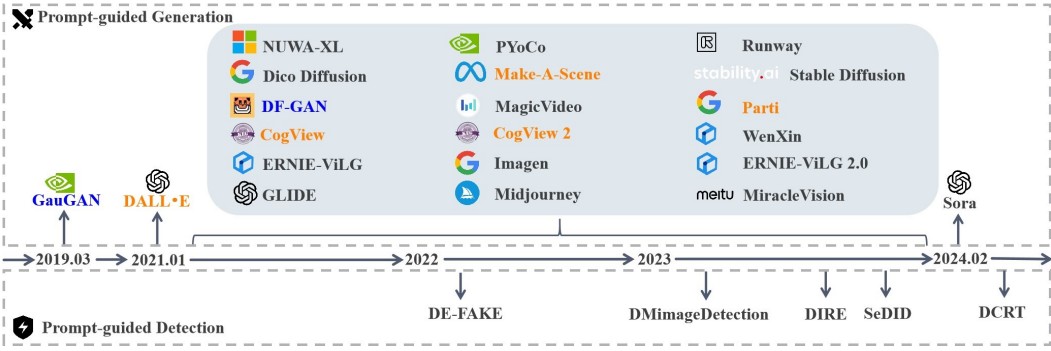

Figure 5: Prompt-guided generation methods/products (above the timeline) and forgery detection techniques (below the timeline) are shown on a chronological timeline. GAN, Autoregressive, and **Diffusion** are marked with blue, orange, and **black** fonts, respectively.

**GAN-based Models.** Based on their model structure, GANs can be classified into single-stage generation networks and stacked architectures. DF-GAN (Tao et al., 2022), a single-stage generation network, uses one generator, one discriminator, and a pre-trained text encoder. It maps text to images by incorporating affine transformations, enabling direct image synthesis from textual descriptions. GoGAN (Mansoor et al., 2022), a stacked architecture, generates higher resolution images in stages. Each branch's generator captures the image distribution, while the discriminator assesses authenticity, refining image resolution and achieving stable training results. Despite their capabilities, GANs face stability issues and mode collapse. These limitations have led to their gradual replacement by autoregressive and diffusion models, which offer improved stability and better handling of diverse data distributions.

**Autoregressive-based Models.** Autoregressive-based models generate images by modeling spatial relationships between pixels and high-level attributes using an Encoder-Decoder architecture with

a multi-head self-attention mechanism. In Text2Image generation, these models convert text and images into token sequences. The autoregressive model predicts image sequences from these tokens, which are then decoded into final images using techniques such as Variational Autoencoders (VAEs) to enhance image quality. Autoregressive models offer explicit density modeling and stable training compared to GANs. Notable examples include DALL·E (Open AI, 2023), which generates creative images from text prompts, CogView (Ding et al., 2021), known for its high-quality image synthesis, and Make-A-Scene (Gafni et al., 2022), which enables interactive image generation. However, autoregressive models face limitations in computational resources, data requirements, and training time due to their large number of parameters. Diffusion models, which offer improved efficiency and require less data, have led to a decline in interest in autoregressive models.

**Diffusion-based Models.** Diffusion-based models have become the state-of-the-art in deep generative models, surpassing previous image and video synthesis techniques. Diffusion models generate images and videos by combining noise prediction models with conditional diffusion or classifier guidance. This process allows the diffusion model to create the desired output based on the provided guidance. These models excel at handling various input conditions and mitigating mode collapse, making them dominant in fields such as Text2Image, Image2Image, Text2Video, and Image2Video synthesis. Notable examples include GLIDE (Nichol et al., 2022), known for its high-quality Text2Image generation; Imagen (Saharia et al., 2022), which excels in photorealistic image synthesis; Sora (Open AI, 2024), a state-of-the-art Text2Video model; and Stable Diffusion (Rombach et al., 2021), which is widely used for its versatility and stability.

### A.3 DETECTION TECHNIQUE FOR TASK-ORIENTED BASED FACE FORGERY

Detection techniques target task-oriented based face forgeries by identifying artifacts left in various feature spaces during the forgery process. These techniques can be categorized into spatial domain-based, frequency domain-based, and temporal domain-based detection technique.

**Spatial Domain-based Detection Technique.** Zhao et al. (2021a) suggests that the key to distinguishing real from forged faces lies in subtle local details. They propose a texture enhancement module, an attention generation module, and a bi-linear attention pooling module to help the model focus on facial texture details. However, these methods often overfit to specific forgery artifacts, leading to a rapid decline in detection performance when faced with unseen forgery methods. To avoid overfitting, researchers have generated forged faces by applying certain operations to real faces. Li et al. (2020a) introduces the FaceX-Ray model, which detects forgery by identifying face fusion boundaries. During training, the model predicts image authenticity and performs pixel-wise classification on the gray scale map of fusion boundaries. This method does not rely on specific forgery artifacts, showing remarkable generalization capabilities in detecting forgeries from unseen methods. Shiohara & Yamasaki (2022) argues that forgeries often contain general forgery traces. They propose Self-Blended Images (SBI), synthetic forgeries created by transforming key points within the same face image, which show strong generalization against unknown forgery methods. However, this method performs poorly against prompt-guidede synthesis methods due to its reliance on the self-forgery process. Cao et al. (2022a) introduces RECCE, combining reconstruction learning and classification to help the model learn compact features of real faces and uncover essential differences between real and fake faces. Some studies have explored the interpretability of deep face forgery detection models. Dong et al. (2022) hypothesizes that detection models identify authenticity by discerning information unrelated to facial identity. They use facial identity as an auxiliary label and designed source feature encoders and target encoders for identity recognition tasks.

**Frequency Domain-based Detection Technique.** Videos and images disseminated across online streaming media often undergo multiple compressions, resulting in low-quality images that obscure forgery artifacts. To address this issue, researchers have explored detection clues in the frequency domain. For instance, Qian et al. (2020a) finds that forgery artifacts can be effectively extracted in the frequency domain. They design a frequency-aware decomposition module to adaptively capture forgery clues within images. Additionally, they introduce a local frequency information statistics module to gather frequency information from each local region of an image and recombine these statistics into multi-channel feature maps for the frequency domain. Since artifacts appear in different regions of various images, Wang et al. (2022) introduces a multi-modal and multi-scale autoregressive model (M2TR) to detect local artifact details at different spatial levels. This model incorporates frequency domain features as auxiliary information, enhancing its capability to detect forgeries in

highly compressed images. While frequency domain-based methods show strong forgery detection capabilities in highly compressed images, their performance significantly declines when encountering unknown forgery methods.

**Temporal Domain-based Detection Technique.** Temporal domain forgery detection focuses on identifying dynamic inconsistencies between video frames over time. Masi et al. (2020) proposes a dual-stream branch network. One branch extracts dynamic temporal inconsistencies from consecutive video frames, and the other amplifies artifact details using a Laplacian of Gaussian (LoG) operator. Recognizing the correlation between forgery and anomaly detection tasks, Ruff et al. (2018) introduces the deep support vector data description (Deep SVDD) loss function to improve the intra-class compactness of real faces and the inter-class distinction between real and forged faces, enhancing the model's generalization capability. Zheng et al. (2021) finds that setting the temporal convolution kernel size to 1 in 3D convolutional kernels enhances the network's ability to capture temporal inconsistencies in forged videos. However, temporal inconsistencies can be compromised by noise, compression, and other factors, leading to reduced robustness in these methods.

### A.4 Detection Technique for Prompt-guided Generation Based Face Forgery

Research achievements in the detection of prompt-guided generation based face forgery are currently limited. Researchers are attempting to break through the mindset of searching for clues specific to task-oriented based face forgery and instead seek the unique fingerprints produced by the prompt-guided generation based face forgery process.

Sha et al. (2023) systematically studies the detection and attribution of fake images generated by diffusion models. They compare the results of image-only input and mixed input (images and corresponding text descriptions) to explore the detection and tracing capabilities of CNN classification models. Corvi et al. (2023) analyzes the frequency domain and model identification capabilities, concluding that diffusion-generated images have unique fingerprints similar to GAN images. Wang et al. (2023b) find that the diffusion reconstruction effect of fake images is superior to that of real images. They use the difference between the reconstructed image and the original image, called Diffusion Reconstruction Error (DIRE), for binary classification to determine authenticity, showing higher generalization ability. Based on this, Ma et al. (2023)and Chen et al. (2024) refine the loss construction of DIRE. However, these methods are tested on small, self-created datasets, and their experimental conclusions lack generality. Additionally, they do not specifically focus on detecting face forgeries. Currently, the detection of faces generated by diffusion models remains relatively unexplored.

## B DeepFaceGen Detailed Statistical Data

In order to construct a robust and extensive benchmark for the detection of face forgery, we carefully consider a range of critical factors including the manner of generation, generation framework, content diversity, ethnic fairness, and label richness throughout the benchmark development process. Following this, we provide detailed introduction to the forged face samples and authentic face samples in DeepFaceGen.

**Forged Face Samples**. The forged face samples of DeepFaceGen consists of 34 types of forgery methods. The number of forged images/videos reaches $350, 264/423, 548$. For content diversity, we collected $143, 579$ forged images and $93, 497$ forged videos from Li et al. (2020b) and He et al. (2021). As shown in Figure 6, the forged images contain 27 forgery methods, including task-oriented based and prompt-guided based generation. Forged samples between both generation methods are roughly balanced. The task-oriented based samples include face swapping, face reenactment and face alteration. In the prompt-guided based generation, sufficient Text2Image and Image2Image samples are generated according to the input modality. At the video-level, a rough balance is similarly maintained between the samples generated by the 16 forgery methods. In the process of generating forged video/image samples, in order to maintain ethnic fairness, we control the balance of skin color through text prompt in prompt-guided based generation. Task-oriented based samples also fit ethnic fairness by employing SkinToneClassifier (Pia & Ma, 2023). Additionally, we employ YOLO (ultralytics, 2020) with manual screening to eliminate low-quality data. The detailed forged statistical data can be seen in Table 2.

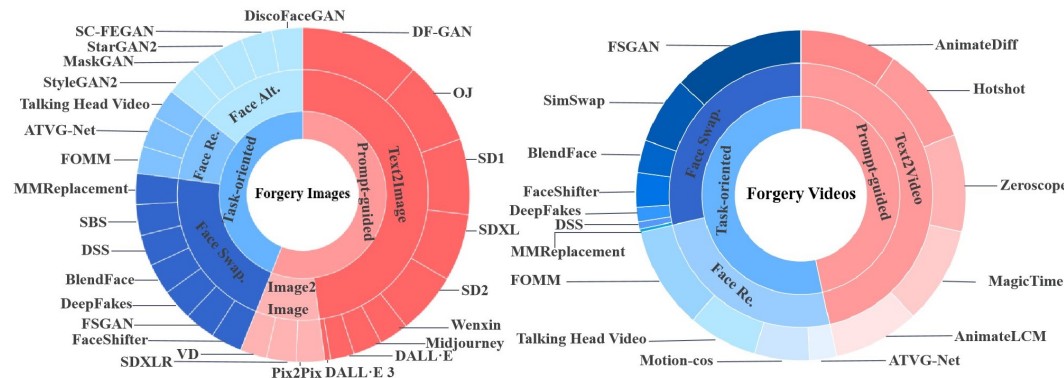

Figure 6: Composition and porportion illustration of image- and video-level sets. At the image-level, DeepFaceGen utilizes 27 face forgery methods. At the video-level, it employs 16 methods. In both levels, the forged data maintains an approximate balance between task-oriented based face forgery technology and prompt-guided generation based face forgery technology.

**Authentic Face samples.** In order to ensure content diversity and ethnic fairness in the authentic face samples used in DeepFaceGen, we obtained real samples from reputable sources including Li et al. (2020b), He et al. (2021), Chen et al. (2023), and Zhao et al. (2019). Specifically, we collected $482$ and $463,101$ real images from Li et al. (2020b) and He et al. (2021), and $19,942,590,99,630,193,245$ real videos from Zhao et al. (2019), Li et al. (2020b), He et al. (2021), and Chen et al. (2023). The final collection consists of $463,583$ images and $313,407$ videos, encompassing diverse ages, genders, skin tones, expressions, hair styles, hair colors, backgrounds, dressing styles, and glasses.

## C    DETAILED DESCRIPTIONS OF PROMPTS CONSTRUCTION

In the design of prompts, we strive to achieve both content diversity and fairness, which are accompanied by a strong emphasis on detailed prompt descriptions. Following this, we designed a complete expressive framework for each prompt sentence based on the face information that humans take into account when describing faces. The prompt sentence framework contains 9 description attributes: ages, genders, skin tones, expressions, hair styles, hair colors, backgrounds, dressing styles, and glasses. Each description attribute contains a detailed scenario situation. By iterating through the combination of 9 attributes, we can generate over $40,000$ prompts. This design ensures data balance across the various text attributes. Then, we use LoRA (Hu et al., 2022) to fine-tune the selected pretrained model and generate forged samples fine-tuned with deepfake samples. The detailed pipeline of prompts construction is shown in Figure 7.

## D    EVALUATION DETAILS

In this section, we provide a detailed introduction to the selected forgery detection methods and disclose the implementation details during the experimental process.

### D.1    FORGERY DETECTION MODELS

Following the basic backone used by the 20 forgery detection methods, we introduce the forgery detection methods in detail.

- **MesoNet** (Afchar et al., 2018) is a face forgery detection algorithm based on mid-level information from image noise. This approach effectively addresses the challenges of diminished image noise and the difficulty of distinguishing forged video frames using high-level semantic features. Its shallow architecture enhances sensitivity to medium and large-scale features, thereby improving the capability of detecting facial characteristics.

Table 2: Detailed Statistical Data of DeepFaceGen.

| Manner | Subset | Methods | Images | Videos | Labels |
|---|---|---|---|---|---|
| Task-oriented | Face Swapping | FaceShifter | 10,500 | 14,387 | n-way labels |
| | | FSGAN | 10,500 | 55,205 | |
| | | DeepFakes | 10,500 | 6,000 | |
| | | BlendFace | 10,500 | 13,491 | |
| | | DSS | 10,500 | 2,866 | |
| | | SBS | 10,500 | - | |
| | | MMReplacement | 10,500 | 1,461 | |
| | | SimSwap | - | 27,786 | |
| | Face Reenactment | Talking Head Video | 9,203 | 28,935 | n-way labels |
| | | ATVG-Net | 10,500 | 11,273 | |
| | | Motion-cos | - | 22,811 | |
| | | FOMM | 10,235 | 42,411 | |
| | Face Alteration | StyleGAN2 | 10,263 | - | n-way labels |
| | | MaskGAN | 8,613 | - | |
| | | StarGAN2 | 10,500 | - | |
| | | SC-FEGAN | 10,500 | - | |
| | | DiscoFaceGAN | 10,500 | - | |
| Prompt-guided | Text2Image | OJ | 28,203 | - | n-way labels prompt labels |
| | | SD1 | 25,677 | - | |
| | | SD2 | 20,898 | - | |
| | | SDXL | 22,839 | - | |
| | | Wenxin | 9,989 | - | |
| | | Midjourney | 9,784 | - | |
| | | DF-GAN | 40,320 | - | |
| | | DALL·E | 8,000 | - | |
| | | DALL·E 3 | 2,000 | - | |
| | Text2Video | AnimateDiff | - | 40,320 | n-way labels prompt labels |
| | | AnimateLCM | - | 35,642 | |
| | | Hotshot | - | 40,320 | |
| | | Zeroscope | - | 40,320 | |
| | | MagicTime | - | 40,320 | |
| | Image2Image | Pix2Pix | 9,620 | - | n-way labels prompt labels |
| | | SDXLR | 9,990 | - | |
| | | VD | 9,130 | - | |
| **Total** | | | **350,264** | **423,548** | |

- **Xception** (Chollet, 2017) is a convolutional neural network architecture entirely based on depthwise separable convolution layers, simplifies the decoupling of channel correlation and spatial correlation to derive depthwise separable convolutions. This enables efficient extraction of complex features from images and video frames.

- **EfficientNet-B0** (Tan & Le, 2020) is the baseline network of the EfficientNet family, which is developed by leveraging a multi-objective neural architecture search based on mobile inverted bottleneck MBConv Sandler et al. (2018) with squeeze-and-excitation optimization Hu et al. (2018) added to it.

- **F3-Net** (Qian et al., 2020b) utilizes two complementary frequency-aware cues: frequency-aware decomposed image components and local frequency statistics. These cues are deeply explored through a dual-stream collaborative learning framework to detect subtle forgery patterns.

- **RECCE** (Cao et al., 2022b) is a reconstruction and classification learning framework designed to learn common characteristics of real faces by reconstructing face images. It

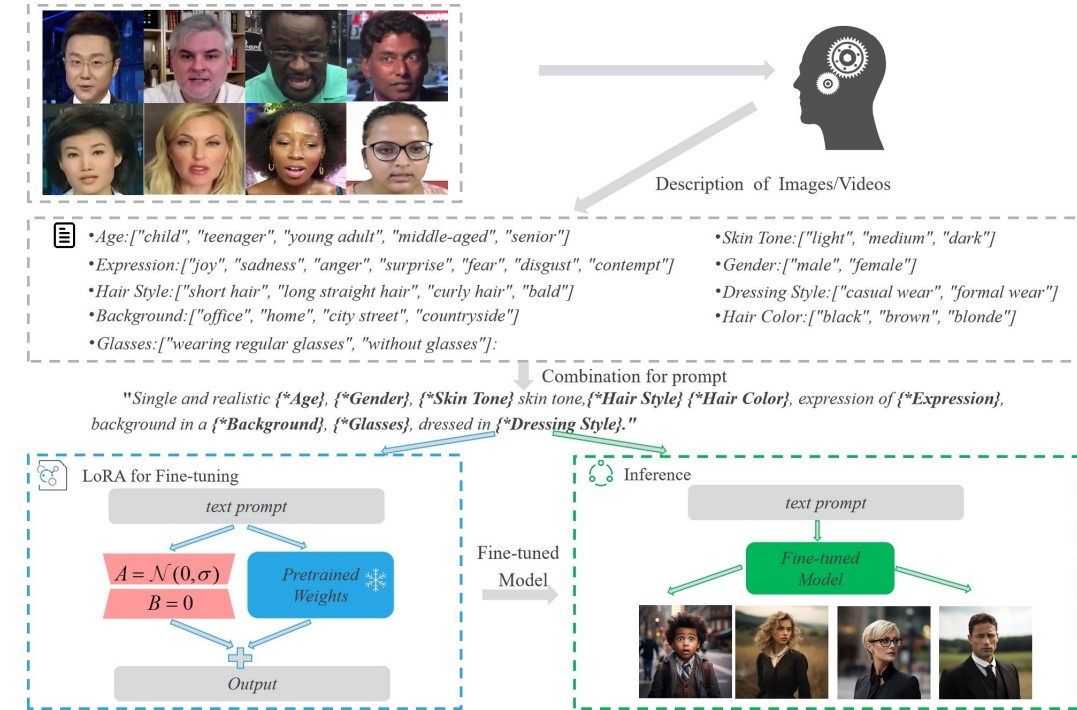

Figure 7: Pipeline of prompts construction. It consists of four parts: the establishment of face description information, the construction of description attributes, the fine-tuning of pre-trained models and the generation of forged samples. After establishing comprehensive attributes to describe face information from images and videos, rich and comprehensive text prompts can be obtained by iterating the combination of description attributes. Then, LoRA (Hu et al., 2022) is used to fine-tune the generative model to the field of face generation for the final generation task.

trains a reconstruction network using real face images and employs the latent features of this network to classify real and forged faces. Due to the inconsistency in data distribution between real and forged faces, the reconstruction errors for forged faces and can accurately highlight the forged regions.

- **DNADet** (Yang et al., 2022) adopts pre-training on image transformation classification and patchwise contrastive learning to capture globally consistent features that are invariant to semantics. It can focus on architecture-related traces and strengthen the global consistency of extracted features.

- **FreqNet** (Tan et al., 2024b) is a lightweight frequency space learning network designed for generalizable forgery image detection. This approach leverages the power of frequency domain learning, providing an adaptable solution for the challenging problem of deepfake detection across diverse sources and GAN models. The methodology includes practical and compact frequency learning plugin modules that integrate with CNN classifiers to enable them to operate effectively within the frequency domain.

- **CViT** (Wodajo & Atnafu, 2021) is a model composed of two main components: Feature Learning (FL) and the Vision Transformer (ViT). The FL component, a stack of convolutional operations without a fully connected layer, extracts features from face images. These features are then processed by the ViT, which converts them into a sequence of image pixels for detection.

- **SLADD** (Chen et al., 2022) aims to generalize well in unseen scenarios. It operates on the principle that a generalizable detector should be sensitive to various types of forgeries. SLADD enriches the diversity of forgeries by synthesizing augmented forgeries using a pool of forgery configurations and enhances sensitivity by training the model to predict these configurations.

- **Exposing** (Ba et al., 2024) is an information bottleneck-based framework for deepfake detection that aims to extract broader forgery clues. It captures a wide range of forgery clues by extracting multiple non-overlapping local representations and fusing them into a global, semantically rich feature.

- **DIRE** (Wang et al., 2023b) is based on the assumption that images generated by diffusion models can be approximately reconstructed through the diffusion process, whereas real images cannot. By applying DDIM's inversion and reconstruction process to the images under inspection, the method differentiates between forged and real samples by analyzing the reconstruction error.

- **DRCT** (Chen et al., 2024) first obtains reconstructed images for both real and fake images based on the diffusion process. It then leverages contrastive learning loss to train a classifier using the four types of images: real, real-reconstructed, fake, and fake-reconstructed. This approach helps establish a more accurate decision boundary for distinguishing between real and fake samples.

- **UnivFD** (Ojha et al., 2023) analyzes the asymmetry in the decision boundary learned by the CNNSpot classifier. While it effectively distinguishes GAN-generated fake images, the feature space of real images lacks independence—i.e., all non-GAN-generated images (real and diffusion-generated images) are classified into a single category. To improve the generalization ability of the detector and enable it to distinguish real from fake images with a balanced decision boundary, a more appropriate feature space is required. To achieve this, Univdf utilizes the pre-trained CLIP model to extract the feature space.

- **NPR** (Tan et al., 2024a) addresses that gap by rethinking CNN-based generator architectures to develop a generalized representation of synthetic artifacts. The research reveals that up-sampling operators, beyond generating frequency-based artifacts, introduce generalized forgery artifacts. Specifically, the local pixel interdependence created by up-sampling in GAN and diffusion-generated images is significant. To capture and characterize these artifacts, the concept of Neighboring Pixel Relationships (NPR) is introduced, providing a new method to identify structural anomalies caused by up-sampling operations.

- **TALL** (Xu et al., 2023) transforms video clips into predefined layouts to preserve both spatial and temporal dependencies, enabling effective detection of Deepfake videos. Specifically, consecutive frames are masked at fixed positions within each frame to enhance generalization performance. These frames are then rearranged into a predefined layout, effectively creating a thumbnail that retains the critical temporal and spatial features for deepfake detection.operations.

- **AltFreezing** (Wang et al., 2023c) identifies that spatial artifacts are more prominent than temporal inconsistencies, leading networks to prioritize learning simpler spatial artifacts. This focus limits the model's ability to leverage all forgery features, ultimately weakening its generalization capacity. To address this, the authors divide the network weights into two groups: spatial-related and temporal-related. During training, they alternate freezing between the two sets of weights, enabling the model to learn both spatial and temporal features effectively. Additionally, a video-level data augmentation method is introduced to further enhance the model's generalization ability.

- **LSDA** (Yan et al., 2024b) tackles the generalization issue in deepfake detection by reducing overfitting to forgery-specific artifacts. It expands the forgery space through variations in the latent space, enabling the model to learn a more generalizable decision boundary. This approach enhances domain-specific features and smoothens transitions between different forgery types, improving cross-domain performance.

### D.2 IMPLEMENTATION DETAILS

**Preproccess.** The image and video datasets are divided into training, validation, and test subsets in a ratio approximately $7 : 1 : 2$. To ensure fairness in evaluation, each subset maintains a ratio of real to fake instances close to $1 : 1$. For video-level evaluations, the video files in the dataset need to be extracted and stored as individual video frames. Given the varying lengths of the video files we collected and generated, we standardize the number of frames extracted from each video to 24. Additionally, since the authors of SLADD (Chen et al., 2022) did not disclose the process

for creating masks, we adopted the following approach: the mask for real data is set to an all-zero matrix, indicating that there are no forgery regions in the input image. For forged data, we use YOLO (ultralytics, 2020) to obtain the face bounding box, and then convert the bounding box into a binary mask image, with the forgery region set to 1 and all other areas set to 0.

**Training.** We all follow the original hyperparameter settings in the evaluation methods. The loss function for SLADD (Chen et al., 2022) is set to MSE, while the loss functions for MesoNet (Afchar et al., 2018), EfficientNet-B0 (Tan & Le, 2020), Xception (Chollet, 2017), F3-Net (Qian et al., 2020b), DNADet (Yang et al., 2022), RECCE (Cao et al., 2022b), and CViT (Wodajo & Atnafu, 2021) are set to CrossEntropyLoss. In particular, based on CrossEntropyLoss, Exposing (Ba et al., 2024) designed the local information loss based on the theoretical analysis of mutual information to ensure the orthogonality and adequacy between local features. The optimizer for all models is Adam with a learning rate of $1 \times 10^{-5}$. The batch size is set to 128. All models are pre-trained on ImageNet. All images in the dataset were resized to a fixed resolution of $299 \times 299$ pixels and normalized to have pixel values in the range [0, 1].

**Inference.** We only perform single-crop inference, and directly scale the input face image to the input spatial size of the model.

## E    DETAILS ON DETAIL EXTRACTION MODULE

*Finding 1* and *Finding 5* indicate that the extraction of detailed features plays a crucial role in detecting both face video and face image forgeries. In this section, we first provide a forward-looking overview of the handling of detailed features within the deepfake detection domain. Following this, we conduct an in-depth analysis through multi-frequency feature analysis, texture feature analysis, and multi-feature fusion experiments. We hope these new insights will offer valuable directions for future research in forgery detection.

As described in Appendix A.3 and A.4, current face forgery detection methods can be categorized into three main types: Spatial Domain-based Detection Techniques, Frequency Domain-based Detection Techniques, and Temporal Domain-based Detection Techniques. Although existing detection methods for prompt-guided generation primarily focus on loss function construction centered around the diffusion process, their core approach still relies on reconstruction error from the input image, placing them within the category of Spatial Domain-based Detection Techniques. Within these three categories, forgery detection methods based on detailed features can be further classified into frequency domain analysis methods (Qian et al., 2020a; Wang et al., 2022), texture feature analysis methods (Zhao et al., 2021b), pixel correlation analysis methods (Tan et al., 2024a; Yan et al., 2024a; Zhong et al., 2024), and pre-trained model feature extraction methods (Ojha et al., 2023).

Given the current state of research, we conduct an in-depth analysis through multi-frequency feature analysis, texture feature analysis, and multi-feature fusion experiments. We hope the conclusions from these experiments will provide valuable foundational knowledge for future research, fostering deeper insights and exploration.

**Multi-frequency Feature Analysis.** We began by applying the Fourier transform to convert the images from the spatial domain to the frequency domain, allowing us to isolate the low, mid, and high-level frequency components using filters. We then performed an inverse Fourier transform to convert the filtered frequency-domain images back to the spatial domain, enabling us to visualize the effects of the filtering. Finally, we trained and tested the NPR (Tan et al., 2024a), Xception (Chollet, 2017), and UnivFD (Ojha et al., 2023) using the visualized low, mid, and high-frequency images. By comparing the detection performance across these frequency bands, we assessed their respective roles in face forgery detection.

As shown in the Table 3, utilizing features extracted from different frequency domains as inputs significantly enhances model performance compared to using the original images alone. *Mid-frequency features perform better in detecting Prompt-guided data, while high-frequency features are more effective for Task-oriented data (Finding 11)*. This is because Task-oriented methods often introduce subtle texture differences or edge inconsistencies, which high-frequency features are adept at capturing. Although mid-frequency features are less detailed in texture extraction, they excel in identifying artifacts from full-image generation in Prompt-guided data. In contrast, low-frequency

Table 3: The ACC of Multi-frequency Feature Analysis. Face Sw., Face Re., Face Al., T2I, and I2I methods are Face Swapping, Face Reenactment, Face Alteration, Text2Image, and Image2Image.

| Detection Feature | Detection Method | Task-oriented | | | Prompt-guided | | Average |
|---|---|---|---|---|---|---|---|
| | | Face Sw. | Face Re. | Face Al. | T2I | I2I | ACC |
| Original | Xception | 65.11 | 62.95 | 58.38 | 73.86 | 69.87 | 66.03 |
| | NPR | 79.51 | 77.32 | 75.56 | 84.02 | 81.65 | 79.61 |
| | UnivFD | 78.41 | 75.02 | 74.65 | 81.56 | 80.01 | 77.93 |
| Low-level | Xception | 64.98 | 63.01 | 59.07 | 74.11 | 70.63 | 66.36 |
| | NPR | 79.66 | 77.4 | 74.98 | 83.99 | 83.65 | 79.93 |
| | UnivFD | 78.08 | 75.42 | 74.37 | 82.01 | 80.22 | 78.02 |
| Mid-level | Xception | 67.52 | 65.01 | 63.98 | 79.36 | 77.01 | 70.57 |
| | NPR | 80.54 | 78.01 | 74.57 | 84.21 | 85.01 | 80.46 |
| | UnivFD | 78.77 | 74.98 | 74.77 | 83.78 | 82.09 | 78.87 |
| High-level | Xception | 69.54 | 68.44 | 67.43 | 75.01 | 72.39 | 70.56 |
| | NPR | 80.77 | 78.64 | 75.01 | 83.71 | 83.87 | 80.40 |
| | UnivFD | 78.89 | 75.48 | 75.21 | 82.99 | 81.07 | 78.73 |

Table 4: The ACC of Texture Feature Analysis. Face Sw., Face Re., Face Al., T2I, and I2I methods are Face Swapping, Face Reenactment, Face Alteration, Text2Image, and Image2Image.

| Detection Feature | Detection Method | Task-oriented | | | Prompt-guided | | Average |
|---|---|---|---|---|---|---|---|
| | | Face Sw. | Face Re. | Face Al. | T2I | I2I | ACC |
| Original | Xception | 65.11 | 62.95 | 58.38 | 73.86 | 69.87 | 66.03 |
| | NPR | 79.51 | 77.32 | 75.56 | 84.02 | 81.65 | 79.61 |
| | UnivFD | 78.41 | 75.02 | 74.65 | 81.56 | 80.01 | 77.93 |
| LBP | Xception | 67.63 | 64.07 | 58.64 | 76.01 | 73.98 | 68.06 |
| | NPR | 79.53 | 77.39 | 75.98 | 84.56 | 82.01 | 79.89 |
| | UnivFD | 78.99 | 75.64 | 74.89 | 81.67 | 78.57 | 77.95 |
| Gabor | Xception | 68.72 | 66.39 | 62.84 | 75.63 | 72.47 | 69.21 |
| | NPR | 80.45 | 78.98 | 77.56 | 84.13 | 81.55 | 80.53 |
| | UnivFD | 79.45 | 76.11 | 76.56 | 82.56 | 80.98 | 79.13 |

features, which capture rough outlines, offer minimal improvement in detection performance when dealing with the high-quality forged data in deepfacegen.

**Texture Feature Analysis.** In the texture feature analysis experiment, we extracted texture features using both Gabor filters and LBP encoding, visualized these features, and used them as inputs for the Xception model for subsequent causal analysis based on the experimental results. The findings, as shown in Table 4, indicate that *texture features enhance the effectiveness of face forgery detection (Finding 12)*. Specifically, Gabor filters, with their sensitivity to image texture features across different orientations and frequencies, are effective at capturing edge and texture variations, making them well-suited for detecting Task-oriented forgery methods. On the other hand, LBP encoding is more inclined to capture global texture patterns, reflecting the overall texture distribution of the image.

**Multi-feature Fusion.** Based on the findings from the Multi-frequency Feature Analysis and Texture Feature Analysis, we explored the potential benefits of Multi-feature Fusion to further enhance detection performance. Specifically, we selected features that demonstrated significant advantages in handling specific categories of data in the previous analyses. We then conducted experiments by concatenating these features for further analysis. The results, as shown in Table 5, indicate that the combination of Gabor Filter and High Frequency features yielded the best performance.

Table 5: The ACC of Multi-feature Fusion. Face Sw., Face Re., Face Al., T2I, and I2I methods are Face Swapping, Face Reenactment, Face Alteration, Text2Image, and Image2Image.

| Texture Feature | Frequency Level | Detection Method | Task-oriented | | | Prompt-guided | | Average |
| --- | --- | --- | --- | --- | --- | --- | --- | --- |
| | | | Face Sw. | Face Re. | Face Al. | T2I | I2I | ACC |
| LBP | Mid | Xception | 66.47 | 67.14 | 62.78 | 81.65 | 80.49 | 71.70 |
| | | NPR | 80.57 | 78.26 | 75.27 | 85.29 | 85.16 | 80.91 |
| | | UnivFD | 78.84 | 75.01 | 74.62 | 84.36 | 83.69 | 79.30 |
| LBP | High | Xception | 69.47 | 69.77 | 65.01 | 75.64 | 76.01 | 71.16 |
| | | NPR | 80.63 | 78.98 | 74.62 | 84.01 | 84.97 | 80.64 |
| | | UnivFD | 78.79 | 75.01 | 75.43 | 83.76 | 82.54 | 79.10 |
| Gabor | High | Xception | 71.65 | 73.87 | 70.32 | 74.34 | 75.42 | 73.12 |
| | | NPR | 81.02 | 79.69 | 76.49 | 86.26 | 86.63 | 82.01 |
| | | UnivFD | 79.25 | 76.15 | 75.46 | 85.99 | 84.63 | 80.29 |
| Gabor | Mid | Xception | 68.41 | 67.52 | 63.41 | 79.87 | 74.89 | 70.82 |
| | | NPR | 80.12 | 78.63 | 74.23 | 83.13 | 84.26 | 80.07 |
| | | UnivFD | 79.65 | 76.05 | 76.48 | 82.69 | 82.01 | 79.37 |

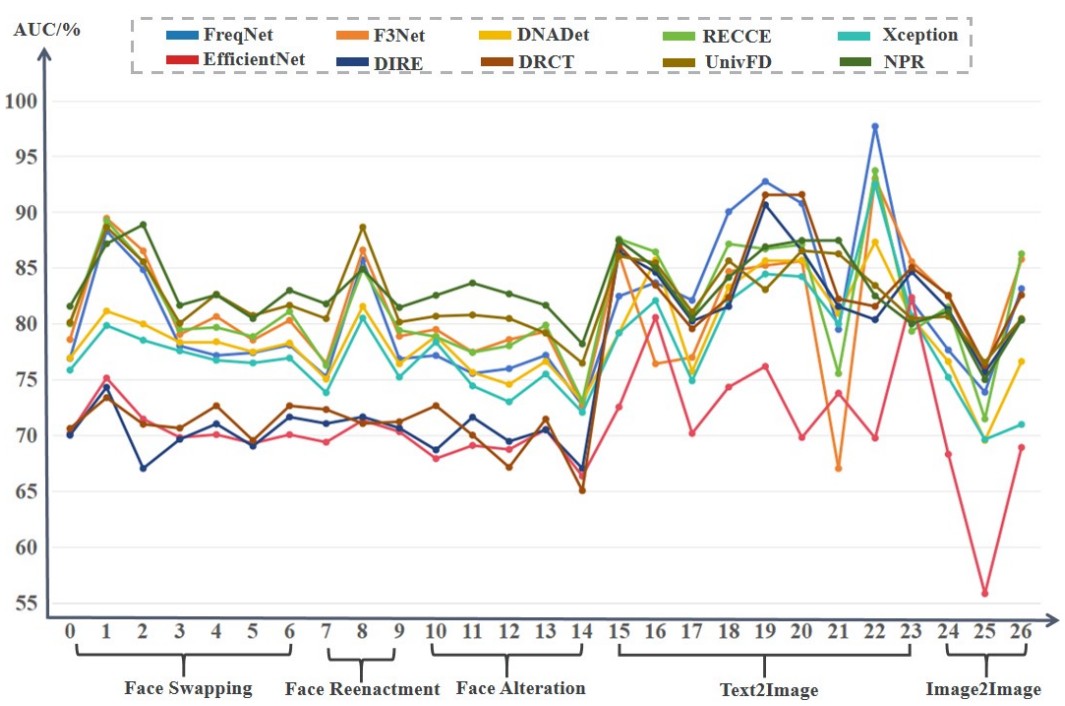

Figure 8: The cross-generalization ability comparison for various image-level forgery detection methods. The horizontal axes represent 5 categories of image forgery techniques. All forgery detection methods are trained on the FaceShifter subset, which has demonstrated the best generalization performance among the detection techniques described in the main manuscript. These methods are subsequently tested using samples generated by the aforementioned forgery techniques.

# F DETAILS FOR CROSS-GENERALIZATION ABILITY VERIFICATION EXPERIMENTS

In this section, we employ 20 forgery detection methods to evaluate the cross-generalization capabilities among sub-datasets. The forgery detection methods are first trained on the subsets that exhibited

Table 6: The AUC scores of Cross-generalization Ability Verification Experiments at image-level (D: Detection technique, F: Forgery method).

| F / D | Year | Modality | MMReplacement | FaceShifter | FSGAN | DeepFakes | BlendFace | SBS | DSS | ATVG-Net | FOMM |
|---|---|---|---|---|---|---|---|---|---|---|---|
| Xception | 2019 | Image | 0.758 | 0.798 | 0.785 | 0.775 | 0.767 | 0.764 | 0.769 | 0.805 | 0.752 |
| EfficientNet-B0 | 2019 | Image | 0.701 | 0.751 | 0.714 | 0.698 | 0.700 | 0.692 | 0.700 | 0.713 | 0.703 |
| F3-Net | 2020 | Image | 0.785 | 0.894 | 0.865 | 0.790 | 0.806 | 0.785 | 0.803 | 0.866 | 0.788 |
| RECCE | 2022 | Image | 0.799 | 0.892 | 0.855 | 0.794 | 0.796 | 0.788 | 0.794 | 0.849 | 0.794 |
| DNADet | 2022 | Image | 0.769 | 0.811 | 0.799 | 0.783 | 0.783 | 0.774 | 0.782 | 0.815 | 0.764 |
| DIRE | 2023 | Image | 0.700 | 0.743 | 0.670 | 0.696 | 0.710 | 0.690 | 0.716 | 0.716 | 0.706 |
| UnivFD | 2023 | Image | 0.801 | 0.886 | 0.855 | 0.800 | 0.826 | 0.807 | 0.816 | 0.886 | 0.801 |
| FreqNet | 2024 | Image | 0.768 | 0.882 | 0.848 | 0.780 | 0.771 | 0.773 | 0.780 | 0.857 | 0.768 |
| DRCT | 2024 | Image | 0.706 | 0.733 | 0.709 | 0.706 | 0.726 | 0.695 | 0.726 | 0.710 | 0.712 |
| NPR | 2024 | Image | 0.815 | 0.871 | 0.888 | 0.816 | 0.825 | 0.804 | 0.829 | 0.848 | 0.814 |

| F / D | Year | Modality | Talking Head Video | StarGAN2 | StyleGAN2 | MaskGAN | SC-FEGAN | DiscoFaceGAN | DALL·E | DALL·E3 | Wenxin |
|---|---|---|---|---|---|---|---|---|---|---|---|
| Xception | 2019 | Image | 0.738 | 0.783 | 0.744 | 0.730 | 0.754 | 0.720 | 0.791 | 0.820 | 0.748 |
| EfficientNet-B0 | 2019 | Image | 0.693 | 0.679 | 0.690 | 0.687 | 0.705 | 0.663 | 0.725 | 0.805 | 0.701 |
| F3-Net | 2020 | Image | 0.764 | 0.794 | 0.773 | 0.785 | 0.792 | 0.727 | 0.862 | 0.764 | 0.769 |
| RECCE | 2022 | Image | 0.762 | 0.788 | 0.774 | 0.780 | 0.798 | 0.730 | 0.875 | 0.864 | 0.803 |
| DNADet | 2022 | Image | 0.750 | 0.788 | 0.756 | 0.745 | 0.766 | 0.729 | 0.791 | 0.857 | 0.757 |
| DIRE | 2023 | Image | 0.710 | 0.686 | 0.716 | 0.694 | 0.704 | 0.670 | 0.864 | 0.846 | 0.802 |
| UnivFD | 2023 | Image | 0.804 | 0.806 | 0.807 | 0.804 | 0.791 | 0.764 | 0.860 | 0.854 | 0.810 |
| FreqNet | 2024 | Image | 0.752 | 0.771 | 0.755 | 0.759 | 0.771 | 0.726 | 0.824 | 0.836 | 0.820 |
| DRCT | 2024 | Image | 0.723 | 0.726 | 0.700 | 0.671 | 0.714 | 0.650 | 0.870 | 0.834 | 0.795 |
| NPR | 2024 | Image | 0.817 | 0.825 | 0.836 | 0.826 | 0.816 | 0.781 | 0.874 | 0.850 | 0.804 |

| F / D | Year | Modality | SD1 | OJ | SD2 | SDXL | DF-GAN | Midjourney | SDXLR | pix2pix | VD |
|---|---|---|---|---|---|---|---|---|---|---|---|
| Xception | 2019 | Image | 0.820 | 0.844 | 0.842 | 0.799 | 0.924 | 0.806 | 0.751 | 0.696 | 0.709 |
| EfficientNet-B0 | 2019 | Image | 0.743 | 0.761 | 0.698 | 0.737 | 0.697 | 0.823 | 0.683 | 0.558 | 0.689 |
| F3-Net | 2020 | Image | 0.846 | 0.852 | 0.856 | 0.670 | 0.930 | 0.855 | 0.824 | 0.758 | 0.857 |
| RECCE | 2022 | Image | 0.871 | 0.866 | 0.870 | 0.755 | 0.937 | 0.793 | 0.815 | 0.714 | 0.862 |
| DNADet | 2022 | Image | 0.832 | 0.856 | 0.856 | 0.809 | 0.873 | 0.805 | 0.766 | 0.695 | 0.766 |
| DIRE | 2023 | Image | 0.815 | 0.906 | 0.865 | 0.815 | 0.803 | 0.846 | 0.812 | 0.756 | 0.803 |
| UnivFD | 2023 | Image | 0.856 | 0.830 | 0.865 | 0.862 | 0.834 | 0.804 | 0.806 | 0.765 | 0.804 |
| FreqNet | 2024 | Image | 0.900 | 0.927 | 0.907 | 0.794 | 0.976 | 0.820 | 0.776 | 0.738 | 0.831 |
| DRCT | 2024 | Image | 0.823 | 0.915 | 0.915 | 0.822 | 0.815 | 0.850 | 0.825 | 0.762 | 0.825 |
| NPR | 2024 | Image | 0.841 | 0.868 | 0.874 | 0.874 | 0.824 | 0.799 | 0.810 | 0.749 | 0.803 |

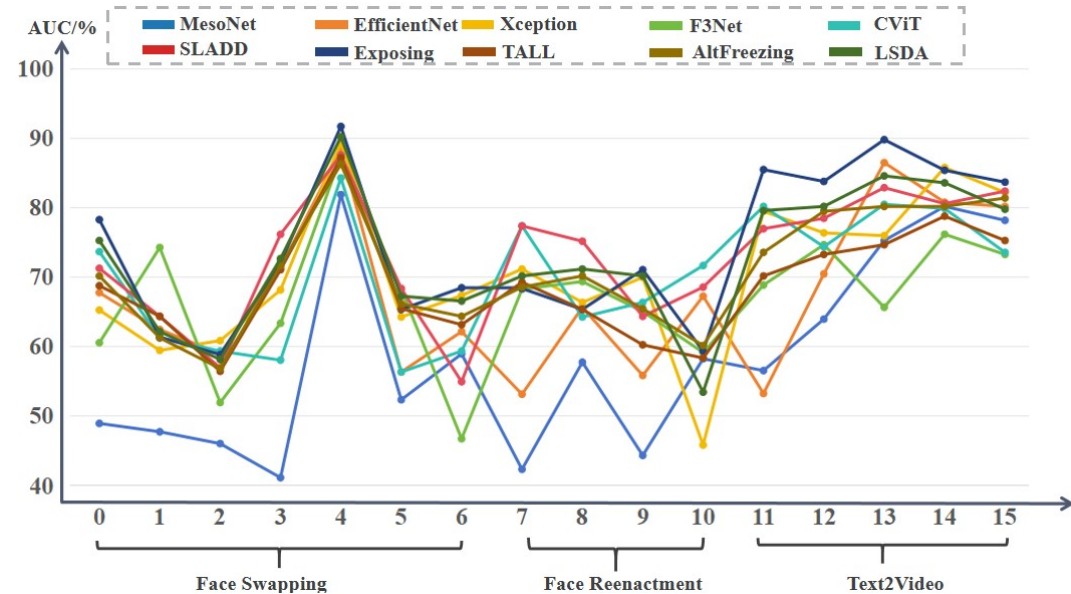

Figure 9: The cross-generalization ability comparison for various video-level forgery detection methods. The horizontal axes represent 3 categories of task-oriented based video forgery techniques. All forgery detection methods are trained on the DSS subset, chosen for its superior generalization performance among the detection techniques described in the main manuscript. Subsequently, these methods are tested using samples generated by the aforementioned video-level forgery techniques.

the best generalization performance in the broad capability evaluation experiments of different forgery techniques discussed in the main text ( FaceShifter subset at the image level and DSS subset at the video level). Subsequently, the generalization performance is tested across various subsets. As shown in Figure 8 and Figure 9, models with detail extraction modules, such as Exposing (Ba et al., 2024), FreqNet (Tan et al., 2024b) and RECCE (Cao et al., 2022b), achieve higher evaluation metrics for identifying editing forged data, which corresponds to *Finding 1*. During the generalization test from task-oriented forgery to prompt-guided generation forgery, it is easier to detect data generated by DF-GAN, further validating *Finding 3*. Additionally, when using task-oriented forgery images/videos as training data, the internal generalization ability of video forgery detection models is significantly lower than that of image forgery detection models, further confirming *Finding 9*. The detailed experimental results can be viewed in Table 6 and 7.

## G  FINE-GRAINED ANALYSIS OF FORGERY DETECTION FEATURE

As shown in Figure 10, we conduct a fine-grained visual analysis of forgery detection features. Based on Figure 10 (a), it is evident that the forgery features of GAN-based model are significantly different from those of Diffusion-based and Autoregressive-based models. This phenomenon provides an explanation for *Finding 3* from the perspective of feature distribution. In Figure 10 (b), the forgery feature distributions are similar when using text and image as input modalities, which corresponds to *Finding 4*. Additionally, Figures 10 (c) and (d) demonstrate that *the forgery features of task-oriented techniques do not show significant differences between images and videos (Finding 13)*.

## H  FINE-GRAINED ATTRIBUTE STATISTIC ANALYSIS FOR DIFFERENT FORGERY TECHNIQUES

In this section, we train all forgery detection models using the training samples obtained from DeepFaceGen. Subsequently, we utilize the fine-grained labels provided by DeepFaceGen to conduct a detailed analysis of the detection patterns of the forgery detection techniques across 9 attributes.

Table 7: The AUC scores of Cross-generalization Ability Verification Experiments at video-level (D: Detection technique, F: Forgery method).

| D \ F | Year | Modality | Talking Head Video | FSGAN | DeepFakes | BlendFace | DSS | MMReplacement |
|---|---|---|---|---|---|---|---|---|
| MesoNet | 2018 | Video | 0.423 | 0.477 | 0.460 | 0.411 | 0.818 | 0.523 |
| EfficientNet-B0 | 2019 | Video | 0.531 | 0.624 | 0.589 | 0.713 | 0.882 | 0.563 |
| Xception | 2019 | Video | 0.711 | 0.594 | 0.608 | 0.681 | 0.893 | 0.642 |
| F3-Net | 2020 | Video | 0.682 | 0.742 | 0.519 | 0.633 | 0.873 | 0.682 |
| CViT | 2021 | Video | 0.773 | 0.612 | 0.593 | 0.580 | 0.842 | 0.563 |
| SLADD | 2022 | Video | 0.773 | 0.643 | 0.569 | 0.761 | 0.875 | 0.683 |
| AltFreezing | 2023 | Video | 0.685 | 0.612 | 0.568 | 0.716 | 0.862 | 0.661 |
| Exposing | 2024 | Video | 0.683 | 0.613 | 0.588 | 0.720 | 0.916 | 0.653 |
| TALL | 2024 | Video | 0.692 | 0.643 | 0.564 | 0.710 | 0.871 | 0.654 |
| LSDA | 2024 | Video | 0.701 | 0.621 | 0.581 | 0.726 | 0.901 | 0.672 |
| D \ F | Year | Modality | SimSwap | FaceShifter | ATVG-Net | Motion-cos | FOMM | AnimateDiff |
| MesoNet | 2018 | Video | 0.589 | 0.489 | 0.577 | 0.443 | 0.582 | 0.565 |
| EfficientNet-B0 | 2019 | Video | 0.621 | 0.677 | 0.656 | 0.558 | 0.672 | 0.532 |
| Xception | 2019 | Video | 0.673 | 0.652 | 0.663 | 0.699 | 0.458 | 0.794 |
| F3-Net | 2020 | Video | 0.467 | 0.605 | 0.693 | 0.650 | 0.591 | 0.688 |
| CViT | 2021 | Video | 0.593 | 0.736 | 0.642 | 0.663 | 0.716 | 0.801 |
| SLADD | 2022 | Video | 0.549 | 0.712 | 0.751 | 0.643 | 0.685 | 0.769 |
| AltFreezing | 2023 | Video | 0.643 | 0.701 | 0.701 | 0.654 | 0.601 | 0.735 |
| Exposing | 2024 | Video | 0.684 | 0.782 | 0.653 | 0.710 | 0.593 | 0.854 |
| TALL | 2024 | Video | 0.631 | 0.687 | 0.653 | 0.602 | 0.583 | 0.701 |
| LSDA | 2024 | Video | 0.665 | 0.752 | 0.711 | 0.701 | 0.534 | 0.795 |
| D \ F | Year | Modality | AnimateLCM | Hotshot | Zeroscope | MagicTime | - | - |
| MesoNet | 2018 | Video | 0.639 | 0.752 | 0.801 | 0.781 | - | - |
| EfficientNet-B0 | 2019 | Video | 0.704 | 0.864 | 0.807 | 0.801 | - | - |
| Xception | 2019 | Video | 0.763 | 0.759 | 0.857 | 0.821 | - | - |
| F3-Net | 2020 | Video | 0.746 | 0.656 | 0.761 | 0.732 | - | - |
| CViT | 2021 | Video | 0.743 | 0.804 | 0.798 | 0.735 | - | - |
| SLADD | 2022 | Video | 0.784 | 0.828 | 0.805 | 0.823 | - | - |
| AltFreezing | 2023 | Video | 0.794 | 0.801 | 0.801 | 0.813 | - | - |
| Exposing | 2024 | Video | 0.837 | 0.897 | 0.853 | 0.836 | - | - |
| TALL | 2024 | Video | 0.732 | 0.746 | 0.787 | 0.752 | - | - |
| LSDA | 2024 | Video | 0.801 | 0.845 | 0.835 | 0.797 | - | - |

**Age Attribute.** The age attribute significantly impacts the effectiveness of forgery detection models. Figures 11 (a) and 12 (a) indicate that forgery detection models face more challenges with detecting forgery samples of children, while it is easier to detect forgery data of elderly faces. This difference is due to the unique facial characteristics of children and the elderly. Children's facial features are finer and smoother, lacking prominent wrinkles and details, which makes it easier for forgery techniques to generate realistic child faces, thereby increasing the difficulty of detection. In contrast, elderly individuals often have more pronounced and complex facial features, including wrinkles, age spots, and sagging skin, which make forgery more challenging and, therefore, more likely to be detected by the model.

**Skin Tone Attribute.** The effectiveness of forgery detection models varies with different skin tones. Figures 11 (b) and 12 (b) show that these models have greater difficulty in accurately detecting forgeries in individuals with darker skin tones compared to those with lighter skin tones. This highlights a racial bias inherent in the forgery detection techniques. The potential cause of this bias could be linked to variations in skin tones and the influence of lighting conditions. Individuals with darker skin tones may have facial features that are harder to capture in forgery detection. Darker skin tones can result in lower contrast in facial details, such as shadows and highlights, making it difficult for forgery detection models to identify forgery artifacts. Conversely, the facial features of individuals with lighter skin tones are generally easier to capture in images. Lighter skin tones make facial details, such as wrinkles and subtle expressions, more visible and typically maintain better facial detail contrast under various lighting conditions.

**Hair Style Attribute.** The variety of people's hairstyles also has an impact on the effectiveness of forgery detection. As shown in Figures 11 (c) and 12 (c), detecting forgeries with the curly hair attribute is more difficult, while detecting those with the bald attribute is easier. In video-level experiments, the detection performance is relatively consistent across different attributes. We infer that curly hair, with its highly complex and irregular structure, contains rich details between strands. This complexity poses a greater challenge for forgery techniques in generating curly hair, making it easier to leave behind subtle artifacts that are difficult to detect. Consequently, detection models

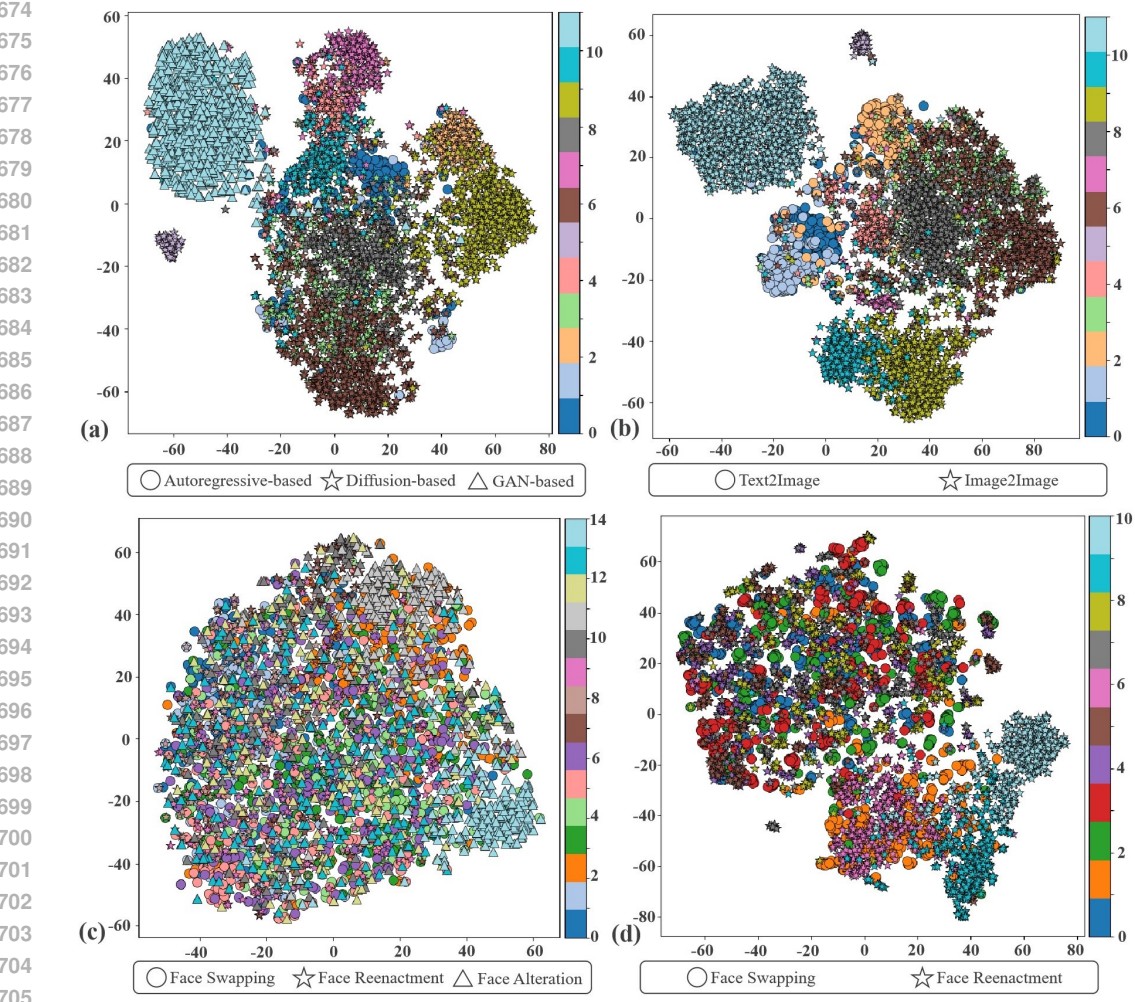

Figure 10: The forgery feature visualization for different forgery techniques on image-level (a-c) and video-level (d) datasets with t-SNE (van der Maaten & Hinton, 2008). (a) different generation frameworks, (b) different input modalities, (c) and (d) different generation manners.

struggle to differentiate these subtle differences, increasing the difficulty of detecting forgeries with curly hair. In contrast, forgery techniques tend to produce more consistent results when generating bald heads due to the lack of complex hair structures, making it easier for detection models to identify forgery artifacts. Additionally, in video-level experiments, the continuity and motion information assist the forgery detection models in capturing forgery artifacts more effectively, leading to more balanced detection performance across different hair style attributes.

**Hair Color Attribute.** Figure 11 (d) and Figure 12 (d) show that forgery detection models perform relatively evenly when detecting forged data with the attributes of brown hair, blonde hair, and black hair. This can be attributed to similar details and contrast under lighting conditions. When generating forged images, forgery techniques typically handle similar textures and lighting effects for all three hair colors. This similarity results in detection models not having significant difficulty differences in identifying these forgeries.

**Expression Attribute.** People's inner emotions can be externalized into different expressions. Based on (e) in Figure 11 and Figure 12, it is apparent that forgery detection models perform well when detecting forged images with the anger and surprise attributes. This may result from the facial expressions of anger and surprise attributes. They contain rich details and features that are easier to extract and recognize in image processing. Tense facial muscles and deep wrinkles are typical

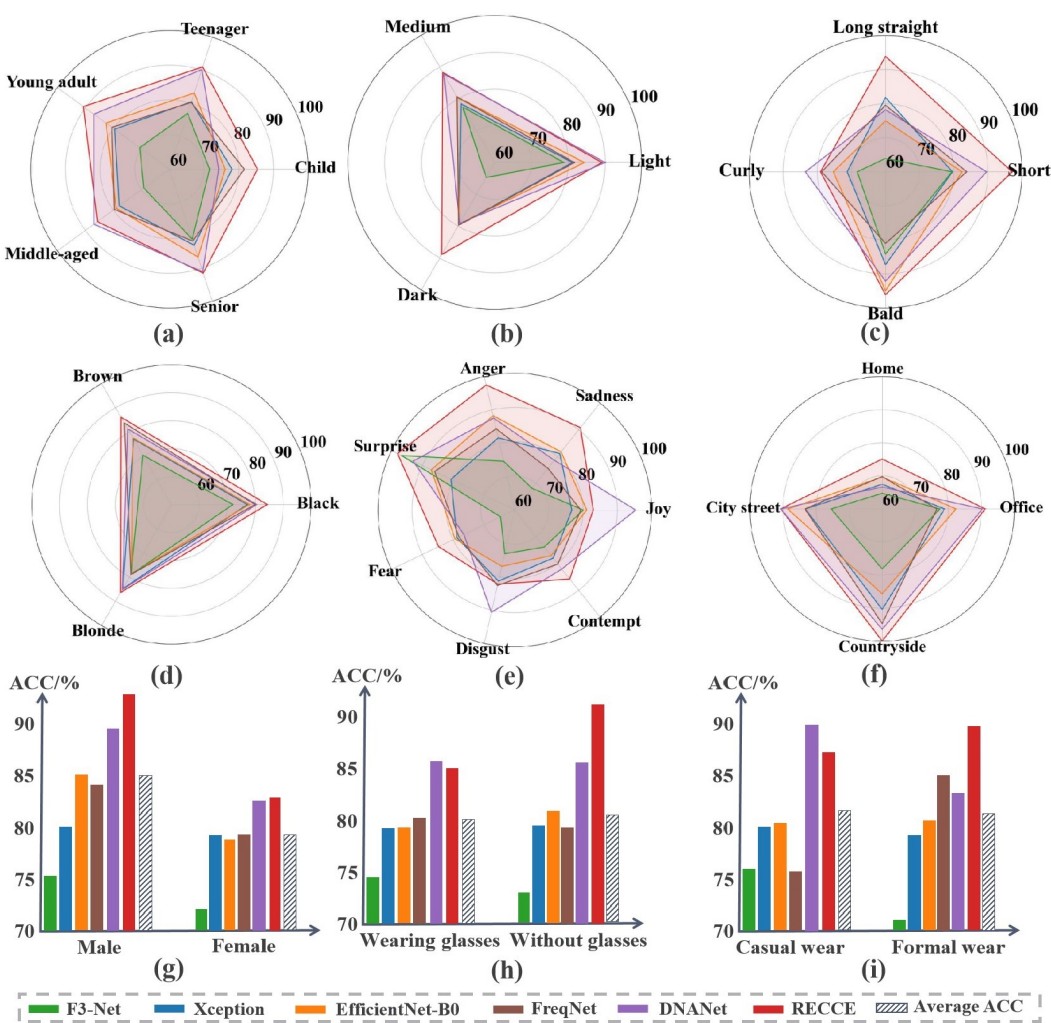

Figure 11: Comparative evaluation of various forgery detection techniques on image-level samples from different attribute perspectives, including (a) age attribute, (b) skin tone attribute, (c) hair style attribute,(d) hair color attribute, (e) expression attribute, (f) background attribute, (g) gender attribute, (h) glasses attribute, and (i) dressing style attribute.

features of anger, while an open mouth and raised eyebrows are clear indicators of surprise. Forgery detection models can use these prominent features to enhance detection accuracy.

**Background Attribute.** The background in images/videos also influences the performance of forgery detection models. Figures 11 (f) and 12 (f) indicate that forgery detection models find it easier to detect forged images with the countryside attribute and harder to detect those with the home attribute. Background complexity may be a direct factor. Countryside backgrounds generally have lower complexity, featuring large natural landscapes such as fields, trees, and skies. These elements are relatively simple and have fewer variations, making it easier for forgery techniques to generate these backgrounds without introducing complex artifacts. Consequently, detection models can more easily identify forged elements in these simple backgrounds. By contrast, home backgrounds typically include many details and complex objects such as furniture, appliances, and decorations. Detection models need to process more details and variations, making it harder to detect forgeries.

**Gender Attribute.** The accuracy of forgery detection models is often lower for female samples ((g) in Figure 11 and 12). Similar to children in age attribute, female facial features are generally finer and smoother, lacking prominent wrinkles and rough skin texture. These fine features may make it

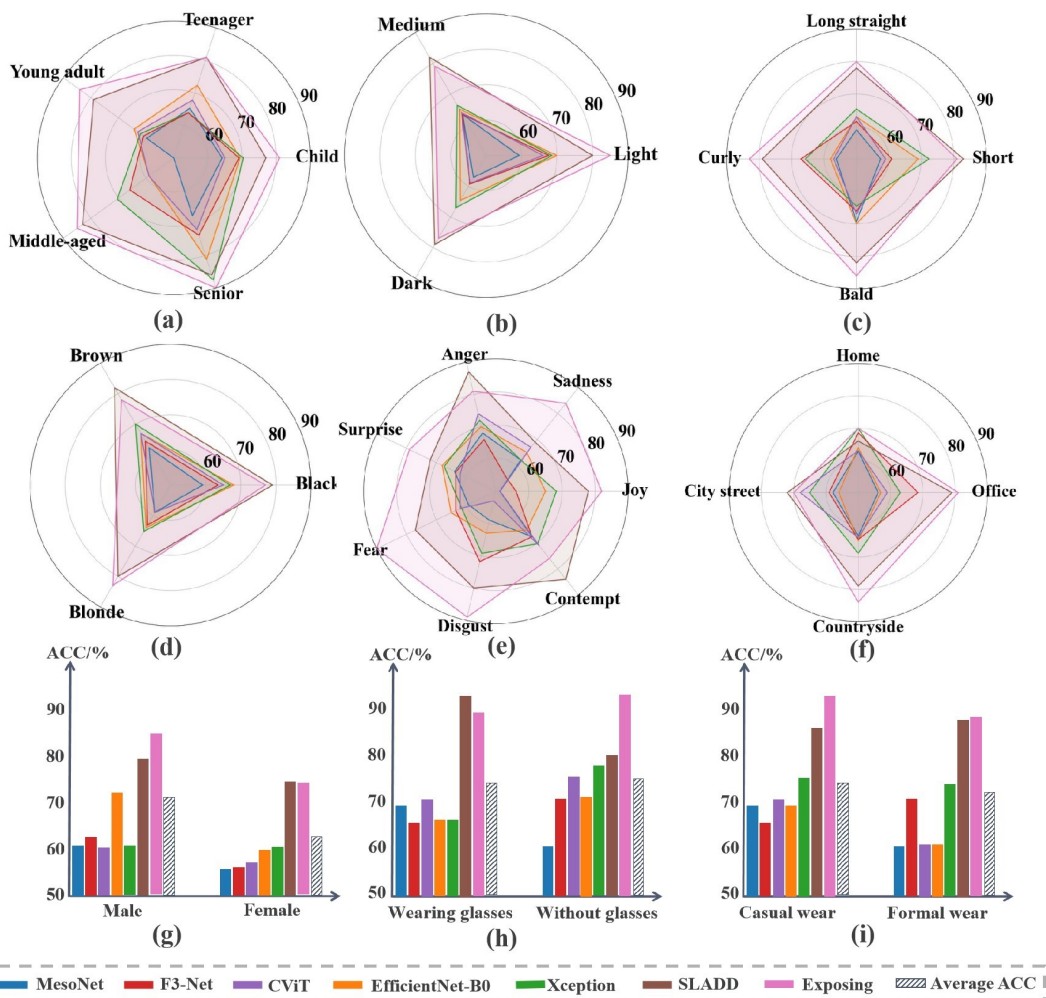

Figure 12: Comparative evaluation of various forgery detection techniques on video-level samples from different attribute perspectives, including (a) age attribute, (b) skin tone attribute, (c) hair style attribute,(d) hair color attribute, (e) expression attribute, (f) background attribute, (g) gender attribute, (h) glasses attribute, and (i) dressing style attribute.

harder for detection models to capture forgery artifacts. Additionally, women tend to wear makeup in greater numbers than men. Cosmetics can enhance or conceal certain facial features, and introduce artificial details such as eyeliner and lipstick. These changes can also make it more challenging for forgery detection models to distinguish between real and forged images, as the makeup may mask subtle forgery artifacts that the model relies on for detection.

**Glasses Attribute.** Based on Figure 11 (h) and Figure 12 (h), forgery detection models perform similarly when detecting forged data with and without the glasses attribute. This can be attributed to glasses' simple and fixed geometric features (such as frames and lenses). When generating faces with glasses, forgery techniques can maintain the stability of these geometric features well, resulting in forged images of similar quality to those without glasses.

**Dressing Style Attribute.** It can be found from Figure 11 (i) and Figure 12 (i) that forgery detection models perform similarly when detecting forged data with the casual wear attribute and the formal wear attribute. This may due to their similar complexity. Although casual and formal wear differ in style, the complexity of details in both types of clothing is relatively similar. Formal wear may include more details (such as ties and buttons), but these details do not significantly affect the quality

of forged images. Casual wear may have more varied styles, but its complexity is comparable to formal wear.

# I    CHALLENGES AND FUTURE WORK

In light of the rapid advancements in face generation techniques, the progress of face forgery detection techniques has significantly lagged behind. Extensive experimentation and analysis reveal several deficiencies in the current forgery detection methods, including inadequate identification accuracy, limited generalization capabilities, and restricted scope for detecting various types of forgery. This section provides a comprehensive overview of the existing challenges in face forgery detection and offers potential valuable directions for future research.

## I.1    CHALLENGES

- **Difficulty in Handling Complex Scenarios.** The diversity of complex scenarios increases the difficulty of face forgery detection tasks. Real-world face forgery detection can be affected by environmental factors such as changes in lighting conditions, which can alter shadows and highlights on the face, making it appear darker or brighter. Changes in camera angles can distort facial shapes and features, making the face look twisted or misaligned. Additionally, variations in background complexity can blur the edges of the face or blend it with the background, making it appear unclear or disproportionate. These factors can impact the authenticity and reliability of detection results, increasing the difficulty of recognizing and detecting forgeries.

- **Poor Generalization Performance.** Although current detection models perform well on individual face forgery datasets, their generalization across different datasets remains inadequate. In real-world scenarios, the type of face forgery method used is often unknown, making it difficult to determine the specific type of forgery. Therefore, using pre-trained face forgery detection models for real-world tasks may result in unreliable detection outcomes.

- **Oversimplified Forgery Detection Tasks.** Current face forgery detection tasks focus primarily on binary classification of whether the content is forged, which is relatively crude. In real-world scenarios, there is often a need for tracing the source of the forgery, which is crucial for determining responsibility and uncovering the truth. In face video forgery tasks, attackers often target only a few video frames or audio segments to alter the video content. However, forgery detection models that focus on video-level forgery detection can easily overlook the characteristics of forged segments, significantly increasing the likelihood of detection errors.

## I.2    FUTURE WORK

- **Objective Quantification of Evaluation Benchmarks.** With the increasingly complex and realistic content forgery scenarios brought about by the development of AIGC technologies, current evaluation benchmarks rely on specific model performance metrics, which can be limiting. In real-world scenarios, designing evaluation benchmarks that can accurately quantify the multi-angle forgery detection capabilities and even the adaptability of models is a crucial direction for future exploration.

- **Dynamic Updating of Benchmark Data.** When designing evaluation benchmarks, it is essential to consider the existence of diverse face forgery types. Regularly updating benchmark datasets to include the latest forgery techniques can help the benchmarks stay close to the complex real-world scenarios. Integrating user feedback data can provide new ideas for dynamically updating benchmark datasets. Additionally, as deep forgery technologies continue to evolve, establishing a dynamic labeling mechanism to address new deep forgery techniques and generative models is becoming increasingly important.

- **Building General Forgery Detection Scenarios.** Although we have constructed a general face deep forgery detection dataset that includes both task-oriented based and prompt-guided generation based face forgery techniques, incorporating both image and video modalities, the audio aspect remains a gap. Furthermore, given the relatively unexplored state of detecting face forgeries generated by diffusion methods, designing general forgery detection

techniques based on the inherent differences between real and forged videos, as well as the local feature similarities and model inference paths, is a critical issue that needs to be addressed in the coming years.

- **Emphasis on Robustness of Forgery Detection Models.** The robustness of forgery detection models is key to maintaining stability and reliability in real-world scenarios with complex and variable content. Introducing adversarial samples during training and testing can enhance the robustness of models. However, while adding noise and adversarial samples can improve robustness to some extent, it can also lead to a loss in detection performance. Exploring the inherent characteristics of real samples to identify differences between forged and real samples and developing detection methods that can handle any face forgery product while ensuring detection accuracy is a primary research direction for the future.

- **Self-Evolving Forgery Detection Frameworks.** Forgery techniques and forgery detection techniques are mutually aligned and promote each other. Forgery technologies generally advance faster than forgery detection technologies, leading to significant harm from forged face products to human society. Current forgery detection models and methods rely mainly on researchers analyzing the flaws and weaknesses of forgery technologies and designing corresponding solutions. Developing self-evolving frameworks using adversarial learning mechanisms and reinforcement learning models to drive the autonomous evolution of forgery detection models, thereby improving the ability to quickly respond to various forgery products, is a key research direction for the future.

## J    POTENTIAL NEGATIVE SOCIAL IMPACTS

The creation and use of deepfake datasets, while beneficial for advancing technology, can lead to several negative societal impacts:

- **Misuse of Forgery Methods.** In order to restore the complex forgery scenes in the real scene as much as possible, the forgery methods in the data set are realistic. These forgery methods can be misused to create misleading or harmful content, eroding public trust in media and making it difficult to distinguish between real and fake information.

- **Ethical Concerns.** Due to the transparency of the data set, a large number of face samples in the data set may provide fake resources for illegal personnel. Widespread exposure to deepfakes can lead to public skepticism and paranoia about the authenticity of all digital content.

To mitigate these impacts, we are contemplating controlled access for users and are committed to the dynamic evolution of DeepFaceGen to ensure it remains robust against emerging threats.

