# OpenReview forum: "A Large-scale Universal Evaluation Benchmark For Face Forgery Detection"
_ICLR.cc/2025/Conference — Submitted to ICLR 2025_

### Official Review · Reviewer_herE · 2024-10-30

**Soundness:** 3
**Presentation:** 2
**Contribution:** 3
**Rating:** 6
**Confidence:** 4

**Summary:**

This paper presents DeepFaceGen, a large-scale evaluation benchmark for face forgery detection combining both task-oriented and prompt-guided generation approaches. The dataset comprises 776,990 real and 773,812 forged face samples (images and videos) generated using 34 different forgery techniques. The authors evaluate 20 mainstream forgery detection methods on this benchmark and derive several key findings, including: (1) detail extraction modules play a crucial role in forgery detection performance; (2) autoregressive-based and diffusion-based techniques produce more realistic forged faces than GAN-based approaches; (3) task-oriented forgeries show better generalization capability than prompt-guided ones.

**Strengths:**

1. The dataset construction demonstrates substantial effort, integrating both task-oriented and prompt-guided forgery samples, while considering racial fairness and content diversity.

2. The experimental analysis reveals several noteworthy findings through comparative studies of detection methods, particularly in understanding the differences between generation frameworks and their generalization capabilities.

3. The paper provides comprehensive technical content, including attribute-level analysis and clear experimental protocols, making it a useful reference for future research in face forgery detection.

**Weaknesses:**

1. The presentation of experimental results lacks clarity, particularly in crucial Figures 1, 2, and 3, which appear cluttered and fail to effectively highlight key findings. The use of small relative bar charts without absolute performance numbers significantly limits the paper's utility as a benchmark study.

2. The claimed findings lack rigorous support and appear overly definitive. Many conclusions are drawn from basic performance comparisons without considering the multiple factors that could influence detection performance. The authors should be more cautious in their assertions and provide stronger empirical evidence for each finding.

3. The crucial issue of generalization capability is inadequately addressed. Figure 3's visualization of cross-generalization results is difficult to interpret, and the corresponding tables in Appendix F are poorly organized. Additionally, the paper lacks comparison with methods specifically designed for generalization, instead focusing mainly on in-domain detection approaches.

4. Section 4.3 and Figure 4's feature visualization analysis is unclear, with the figure being particularly difficult to distinguish. It's not evident how this analysis contributes to the paper's overall findings.

Overall, while the paper attempts to cover extensive ground, it suffers from imprecise presentation and insufficiently supported conclusions. A substantial revision is needed to improve the result presentation, strengthen the empirical support for findings, and make the content more accessible to readers.

**Questions:**

See the Weaknesses section

---

> ### Author Response · Authors · 2024-11-23
> **Rebuttal by Authors**
>
> Thanks for thoughtful and constructive feedback. We are glad to hear that you found the DeepFaceGen benchmark valuable. Below are responses to each of your comments. (The anonymous website referred to below is https://anonymous.4open.science/r/herE-CF34).
>
> ***W1: a) Figures 1-3 lack clarity which fail to effectively highlight key findings;  b) The use of small relative bar charts limits utility.***
>
> a) Thanks for valuable advice. Figures 1-3 present the overall performance on the benchmark. To better highlight findings, we will break these images into subfigures aligned with specific observations of each finding in the final version. For example, Finding 3, which examines detection difficulty based on the generation framework (autoregressive, diffusion, and GAN), will have separate performance plots for each category to better illustrate the underlying patterns.
>
> b) Thank you for your valuable feedback. We will provide a table summarizing the detailed overall scores of the 20 detection models from Figures 1 and 2. Additionally, we have included tables showing the detailed performance of different forgery detection methods on various forgery types. Please refer to Section 1 of the anonymous website for the results.
>
> ***W2: Findings lack rigorous support. Should be more cautious in their assertions and provide stronger empirical evidence.***
>
> Thanks for your insightful feedback. To validate the effectiveness of detail extraction modules, which are their commonalities across SOTA models performing well on both task-oriented and prompt-guided data, we conducted experiments involving multi-frequency feature analysis, multi-texture feature analysis, and multi-feature fusion.
>
> Furthermore, we are actively working to provide more robust evidence for our experimental findings. Specifically, we have systematically reviewed all findings, revealing that ***the discoveries in DeepFaceGen stem from the inherent differences between task-oriented and prompt-guided data, as well as the unique challenges models face when detecting them***. To explore this further, we conducted visual analyses across three dimensions: ***frequency domain, noise domain, and spatial domain***. These analyses highlight distinct data characteristics and the varying effectiveness of detection strategies, establishing a clear connection between observations and conclusions. For detailed classifications, please refer to Section 2 of the anonymous website.
>
> ***W3: a）The crucial issue of generalization capability is inadequately addressed. Figure 3's visualization of cross-generalization results is difficult to interpret, and the corresponding tables in Appendix F are poorly organized. b）Lack comparison with methods specifically designed for generalization.***
>
> a) Apologize for the difficulty in understanding. Figure 3 shows the results of cross-training and testing using the Xception model on image and video subsets, grouped by forgery method. The redder the block, the higher the metric; the bluer the block, the lower the metric. Appendix F validates the experiments by training 18 additional evaluation methods on the subset where Xception performs best and testing them on other subsets. To better highlight our findings, we will include a table in the final version summarizing generalization performance by forgery category, such as training on text-to-image data and evaluating on face-swapping data.
>
> b) Thanks for valuable suggestions. Building on the generalization experiments within DeepFaceGen, we conducted cross-domain generalization tests.
>
> Specifically, we explored the model's generalization when trained on DeepFaceGen and tested on non-face data (Genimage).  We get the following findings: ***1) Directly applying face AIGC to non-face AIGC presents substantial challenges. 2) Face detection models can adapt to identify non-face features when trained on similar generative frameworks. 3) Models focus primarily on facial forgery artifacts, which hinders their ability to generalize effectively***. Detailed analysis can be found in Section 3 of anonymous website.
>
> ***W4: Section 4.3 and Figure 4's feature visualization analysis is unclear, with the figure being difficult to distinguish.***
>
> Sincerely sorry for the confusion. Section 4.3 aims to visualize the feature distribution using T-SNE to examine whether the model can distinguish task-oriented and prompt-guided forgery features. To clearly show the feature distribution for each forgery type, Figure 4 presents the T-SNE results in the form of a "secondary directory" (refer to Table 2 in the appendix, lines 1189-1226). To better highlight Finding 10, we will include the distribution of task-oriented and prompt-guided types in the final version. For details, please refer to Section 4 on the anonymous website.

---

### Official Review · Reviewer_3GRu · 2024-11-02

**Soundness:** 2
**Presentation:** 3
**Contribution:** 2
**Rating:** 5
**Confidence:** 5

**Summary:**

This work presents a face deepfake dataset that incorporates 34 face forgery generation techniques. The author conducts training and evaluation experiments on this dataset and presents several findings. This author implements both image and video detectors for a more comprehensive comparison.

**Strengths:**

- This work implements 34 distinct face forgery techniques for creating fake data, although this work does not create new real faces in their dataset.
- This work involves both spatial and temporal detection methods for comparison, which is new in the existing face forgery benchmark.
- Some findings are new to me such as Finding-10.

**Weaknesses:**

**1. Unclear and ambiguous definitions for the forgery types:** In this paper, face forgery techniques are categorized into two main types: prompt-guided and task-oriented. However, both of these concepts can be rather abstract and unclear, potentially misleading the community. Specifically:
- In Line 48, the author mentions for the first time that "current deepfake datasets focus on relatively outdated task-oriented based face forgery techniques." However, the meaning of "task-oriented-based face forgery" is not clear. To the best of my knowledge, I have not found any existing research or survey that provides such a definition of face forgery techniques. The author must either provide **appropriate citations and references or clearly define the rationale behind this definition**.
- If the author cannot provide accurate references or detailed explanations, the categorization will remain very unclear and ambiguous. I am not sure whether this category is reasonable and can be accepted by the community.


**2. Several findings in this work have been verified by previous works and might be trivial:**
- Finding-1 and Finding-5 are quite similar, indicating the need to extract more detailed information. In fact, many SOTA detectors have proposed different approaches to enhance detection performance by leveraging more discriminative artifacts or detailed information. So, the conclusion that these SOTA detectors can achieve better performance than the vanilla backbone such as Xception can be rather trivial.
- Finding-11 concludes and highlights the benefit of using different levels of frequency clues for detection. However, **many previous (very early) works have already provided this conclusion**, such as F3Net (ECCV'20), SRM (CVPR'21), and SPSL (CVPR'21). These works reached these conclusions using very old benchmark datasets such as FF++ (ICCV'19). But the author seems to use the latest benchmark and still concludes the same conclusion. What is the significance of creating such a benchmark?
- Finding-6: "existing diffusion-based generation techniques possess a comparable ability to generate forged videos." The same conclusion has been proposed in works like ref[1], making this conclusion trivial.


**3. About the correctness of several findings:**
- In Finding 6, the authors use "detection performance" to infer "the generation quality of diffusion techniques." However, the two may **have a correlation but not a causal relationship**. The worst detection performance can be attributed to many scenarios, while the metrics for quality are FID-like tools used in the image synthesis field. Misconnecting these two concepts is inappropriate. Specifically, I note that diffusion-generated images can have different poses and facial sizes compared to other data, which makes their distribution differ from other deepfake data (as seen in the left-up and right-below corners in Figure 7). But this does not mean that diffusion-generated images are not realistic.


**4. Reproduction Concerns:**
- I have checked the link (`https://anonymous.4open.science/r/DeepFaceGen-47D1`) provided by the author. It seems to only contain the `model.py` files. Without detailed preprocessing, training, and evaluation codes, it is hard to see whether the results are reproducible and the experimental settings are fair.

ref[1]: Diffusion Models Beat GANs on Image Synthesis, NeurIPS'21.

**Questions:**

1. What is the meaning of "prompt-guided" and "task-oriented"? What is the rationale or reference for this definition of forgery types?

2. Which findings presented in this work are truly new to the field and have not been shown in existing or previous works?

3. Several recent face forgery datasets such as those mentioned in ref[1,2,3,4,5] are missing from the discussion. What are the advantages and additional contributions of the current dataset over these previous ones?

4. What are the deeper insights regarding the differences between image and video face forgery detectors?

5. Most of the findings and evaluations are empirical observations. What are the underlying reasons behind these observations?

ref[1]: Diffusion Facial Forgery Detection, ArXiv 2024.

ref[2]: DiffusionFace: Towards a Comprehensive Dataset for Diffusion-Based Face Forgery Analysis, ArXiv 2024.

ref[3]: Contrastive pseudo learning for open-world deepfake attribution, ICCV 2023.

ref[4]: DF40: Toward Next-Generation Deepfake Detection, NeurIPS 2024.

ref[5]: AI-Face: A Million-Scale Demographically Annotated AI-Generated Face Dataset and Fairness Benchmark, ArXiv 2024.

---

> ### Author Response · Authors · 2024-11-23
> **Responses to W1-W4**
>
> Thank you for your thoughtful feedback and for recognizing the strengths of our work. Below are our responses to ***W1-W4***. (The anonymous website referred to below is https://anonymous.4open.science/r/3GRu-D654).
>
> ***W1:Unclear and ambiguous definitions***
>
> Sorry for the confusion. Task-oriented-based face forgery summarizes the commonalities of traditional forgery methods, which are task-specific (e.g., face swapping, face alteration, and face reenactment). In contrast, prompt-guided-based face forgery focuses on the core technology of current AIGC generation techniques, which are driven by prompts (e.g., text2image, image2image, text2video). A detailed explanation and justification of this classification is provided below:
>
> Due to the emergence of new AIGC forgery technologies, there is currently a lack of effective definitions distinguishing between "traditional" and "novel" methods. To clarify, we reviewed benchmarks that aim to differentiate between these two categories, and we found they typically rely on the following classification approaches:
>
> - ***Model frameworks*** (e.g., GAN-based vs. Diffusion-based).
> - ***Forgery regions*** (e.g., full-image generation, face reenactment, face swapping).
> - ***Generation conditions*** (e.g., text2image, image2image, face swapping, face alteration).
>
> For DeepFaceGen, which covers a comprehensive set of forgery methods, ***using the Model Frameworks and Forgery Regions approaches results in overlap between "traditional" and "novel" categories***. Therefore, we ***categorize primarily by generation conditions***. "Traditional" forgery methods focus on specific sub-tasks (e.g., face swapping, face alteration, and face reenactment), so we classify them as "Task-oriented." On the other hand, "novel AIGC" methods, despite differences in forgery region and model framework, are all prompt-driven (e.g., text2image, image2image, text2video), which is central to their free-form generation. Thus, we classify them as "Prompt-guided." The additional dataset you provided, Ref [1], supports this view and avoids overlap between "traditional" and "novel AIGC" data in this classification.
>
> Thank you for your insightful question, which reflects your deep understanding of the field. Your feedback has prompted us to clarify the classification approach in the definition section and provide a detailed explanation in the appendix.
>
> ***W2: a) Finding-1 and Finding-5 can be trivial; b) Previous works have already provided Finding-11. What is the significance of Deepfacegen；c) Finding-6: The same conclusion has been proposed by Ref[2].***
>
> a) Sorry for the confusion. Previous analyses focused on traditional forgery data. However, our findings in Finding-1 and Finding-5 also validate the effectiveness and patterns in novel forgery data, which helps advance the development of new forgery detection technologies in the field.
>
> b) Sorry for the confusion. Finding 11 is derived from both traditional "Task-oriented" and newer "Prompt-guided" data, making it more universal and comprehensive. Previous methods provided only a rough analysis of task-oriented data in the frequency domain. In this work, we extend the analysis to prompt-guided data and uncover more detailed findings: ***mid-frequency features are more effective for detecting prompt-guided data, while high-frequency features perform better for task-oriented data***.
>
> c) Sorry for the misunderstanding. Ref[2] suggests that diffusion-based images outperform those generated by GANs and exhibit similar effects across various generation scenarios. Finding 6 highlights that current diffusion-based forged videos display similar forgery cues during detection.
>
> ***W3: On the Incorrect Presentation of Finding 6***
>
> Thank you for pointing this out. We agree that detection performance doesn’t directly reflect the quality of forged data. Our intent was to show that, based on the performance of video forgery detection models on DeepFaceGen, existing diffusion-based techniques exhibit similar forgery artifacts, even as diffusion models evolve. We appreciate your careful review, and will clarify Finding 6 in the final version to avoid any misunderstandings.
>
> ***W4:Reproduction Concerns***
>
> Thank you for your valuable suggestions. We have provided detailed experimental settings in the Evaluation Settings section and Appendix D.2. Additionally, we have maintained the open-source website to improve user access and usability of DeepFaceGen.
>
> Ref[1]: Diffusion Facial Forgery Detection, ArXiv 2024.
>
> Ref[2]: Diffusion Models Beat GANs on Image Synthesis, NeurIPS'21.

---

> ### Author Response · Authors · 2024-11-23
> **Responses to Q1-Q5**
>
> Thank you for your thoughtful feedback and for recognizing the strengths of our work. Below are our responses to ***Q1-Q5***. (The anonymous website referred to below is https://anonymous.4open.science/r/3GRu-D654).
>
> ***Q1: Meaning and  rationale for forgery type definition.***
>
> Please refer to the answer of ***W1***.
>
> ***Q2: New findings?***
>
> Sorry for the confusion. Unlike previous studies based on traditional forged data, our findings are built on comprehensive evaluations covering both traditional and novel forgery methods. Based on this, we categorize the findings into two levels:
>
> ***Generalization Validation on Novel Prompt-Guided Forgeries***
> - Finding 1 & 5: General findings derived from the commonalities of SOTA detection models.
> - Finding 11, 12 & 13: Insights focused on forgery data detail extraction.
> - Finding 2 & 7: Common forgery cues identified through the performance of detection models.
> - Finding 8, 9 & 10: Generalization findings when models are tested across task-oriented and prompt-guided data.
>
> ***Unique Findings on Prompt-Guided Data***
> - Finding 3 & 6: General findings on detection difficulty across different forgery frameworks and techniques.
> - Finding 4: Insights into the impact of different input modalities on detection difficulty.
> - Finding 14: Findings on how variations in facial attributes affect detection performance.
>
> Detailed results can be found in Section 1 of the anonymous website. (We have also been conducting more in-depth experiments using DeepFaceGen. If interested, please refer to Sections 3-7 on the anonymous website.)
>
> ***Q3: Recent datasets are missing from the discussion.***
>
> Thank you for pointing this out. We will include the dataset comparisons you provided in the RELATED WORKS section of the final version. Overall, the additional datasets you mentioned provide valuable contributions to the community, but they only cover a limited set of generation methods. There are significant limitations in data richness, modality diversity, and data fairness. For more details, please refer to Section 2 of the anonymous website.
>
> ***Q4: Deep insights regarding the differences between image and video face forgery detectors?***
>
> Thanks for your valuable feedback. Image forgery detection focuses on spatial and frequency domain features:
> - ***Spatial Domain***: Analyzes distribution anomalies at the pixel or patch level.
> - ***Frequency Domain***: Identifies common forgery artifacts in forged versus real images.
>
> Video forgery detection, with its richer temporal data, faces greater challenges in feature extraction. It emphasizes ***temporal correlations between frames*** to capture forgery anomalies. Despite their differences, image and video detection offer complementary insights, with many unexplored areas for cross-domain research. Here are our insights:
>
> - ***Incorporating Temporal Information from Image Forgery Detection***: In GAN and diffusion architectures, forgery detection based on reconstruction error has so far treated generative models merely as auxiliary tools.  However, the generation process itself produces temporally varying image sequences.  By delving into the principles of the forgery models, we can extract temporal cues during generation, potentially enhancing image-based detection.
>
> - ***Prompt-guided Forgery Detection in Video***: As highlighted in Appendix A.4, prompt-guided forgery data remains unexplored in video forgery detection. DeepFaceGen fills this gap by providing the necessary data support. Techniques from image forgery detection, such as noise reconstruction and contrastive learning between prompts and generated outputs, can offer valuable insights for detecting prompt-guided video forgeries.
>
> - ***Introducing Graph Structures***: Forgery detection in the field rarely explores graph structures. In other domains, such as spatiotemporal data, satellite navigation, and social network detection, graph structures have proven effective. Human face videos can essentially be viewed as dynamic graph structures, and constructing such structures is crucial for enhancing forgery detection.
>
> ***Q5.Empirical observations. What are the underlying reasons behind observations?***
>
> Thanks for your valuable advice. We conducted detailed experimental analysis of the extraction module characteristics in SOTA models, resulting in Findings 11 and 12 (Appendix E). Furthermore, ***the findings in DeepFaceGen are derived from the differences between task-oriented and prompt-guided data, as well as the varying challenges models face when detecting them***. To explore this, we conducted visual analyses from three perspectives: ***frequency domain, noise, and spatial domain***. These analyses clearly reveal differences in the data itself and the effects of distinct detection approaches, establishing a strong connection between observations and conclusions. Detailed analysis is available in Section 8 of the anonymous website.

---

### Official Review · Reviewer_UKvR · 2024-11-03

**Soundness:** 3
**Presentation:** 3
**Contribution:** 3
**Rating:** 6
**Confidence:** 3

**Summary:**

The paper introduces DeepFaceGen, a benchmark specifically developed for face forgery detection. DeepFaceGen includes 463,583 real images, 313,407 real videos, 350,264 forged images, and 423,548 forged videos, generated using 34 common image/video generation techniques. The authors comprehensively evaluate 20 mainstream face forgery detection techniques using DeepFaceGen.

**Strengths:**

1. The paper constructs a large face forgery dataset, utilizing a variety of generation methods, including Prompt-guided Generation techniques such as Text2Image, Image2Image, and Text2Video, as well as Task-oriented Generation techniques. Compared to previous datasets, this dataset employs a more comprehensive and diverse range of generation methods, resulting in a significantly larger scale.

2. The paper employs 20 detection methods and conducts extensive experiments and analyses using DeepFaceGen.

**Weaknesses:**

1. The paper may not present a methodological contribution in itself, but the large and comprehensive dataset is valuable.
2. I have some questions regarding missing details. The authors selected 20 detection methods, including both image-level and video-level approaches, but did not clarify the representativeness of these methods.  Additionally, while the authors conducted analyses of texture and frequency domain features, the chosen methods seem to focus more on detection models that utilize frequency domain features.

**Questions:**

See weaknesses.

---

> ### Author Response · Authors · 2024-11-23
> **Rebuttal by Authors**
>
> Thank you for your diligent review and comments. We are glad that you appreciated the comprehensiveness of our dataset and your recognition of our extensive experiments. (The anonymous website referred to below is https://anonymous.4open.science/r/UKvR-AE1E).
>
> ***W1: The paper may not present a methodological contribution in itself.***
>
> We sincerely apologize for the confusion.  With DeepFaceGen's comprehensive coverage of forgery data types (task-oriented and prompt-guided), we can expand or refine findings derived from earlier, more limited datasets.  We have also conducted in-depth explorations using DeepFaceGen.  Below are additional findings, which will be included in the final version.
>
> - ***Capturing Temporal Forgery Artifacts in Diffusion-Generated Images***: Introducing temporal information during the denoising process of diffusion improves forgery detection performance.
>
> - ***Investigating the Impact of Identity Information on Face Forgery Detection Performance***: Models that extract information directly from the data are affected by ID information, whereas reconstruction-based methods are not significantly influenced by ID information in their detection performance.
>
> -  ***Cross-domain Detection Performance from Face to Non-face Scenarios***: (1) Directly applying face AIGC to non-face AIGC presents substantial challenges. (2) Face detection models can adapt to identify non-face features when trained on similar generative frameworks. (3) Models focus primarily on facial forgery artifacts, which hinders their ability to generalize effectively.
>
> - ***Exploring Symmetry Differences Between Real and Prompt-Guided Data***：Focusing on symmetry differences could help capture the physiological discrepancies between real and prompt-guided generated data, providing a new approach for image/video face forgery detection.
>
> - ***The impact of various architectures  on forgery detection***：Through extensive comparisons of CNN-based and Transformer-based detection models， we found that in the video forgery detection domain, Transformer-based models generally outperform CNN-based models.
>
> - ***Exploring data augmentation methods for improved detection performance***: We explored the impact of pseudo-blending-fake data on detection performance using DeepFaceGen. Through extensive comparisons of model performance on both domain-specific and cross-dataset tasks with and without the use of pseudo-blending fake samples, we observed consistent improvements in detecting face-swapping forgeries.
>
> Detailed analysis is available in Sections 1-6  of the anonymous website.
>
> ***W2: a) The authors did not clarify the representativeness of 20 detection methods.b) the chosen methods seem to focus more on detection models that utilize frequency domain features.***
>
> a）Sorry for the confusion. In Appendix D (EVALUATION DETAILS), we provide an overview of the 20 evaluation methods, detailing their key characteristics, such as the technical approach and the types of forgery data they are suited for.  We will provide more detailed descriptions in the main body of the final version. A summary table showcasing the representativeness of the 20 detection methods is provided in Section 7 of the anonymous website.
>
> b) Thank you for your valuable advice. Using forgery detection models with different approaches indeed provides more comprehensive and reliable results. We have added F3Net , EfficientNet and RECCE. As a result, the detection methods in the "Details on Extraction Module" now cover naive detector, frequency domain, spatial domain, and pre-trained model-based feature extraction methods. The additional experiments show that using mid-frequency features significantly improves detection for prompt-guided types, while high-frequency features enhance detection for task-oriented types. Furthermore, texture features improve detection performance for both prompt-guided and task-oriented types, consistent with the validation experiments in the main text. More experimental results can be found in Section 8 of the anonymous website.

---

### Official Review · Reviewer_G6No · 2024-11-03

**Soundness:** 3
**Presentation:** 3
**Contribution:** 3
**Rating:** 5
**Confidence:** 5

**Summary:**

This work proposes a large-scale benchmark for the deepfake detection task, and the authors draw some findings through extensive experiments.

**Strengths:**

This paper is easy-to-follow. This work discusses large-scale deepfakes generated by a variety of methods, including recent prompt-guided deepfakes, which is a limitation of prior datasets.
Since deepfake media are rapidly evolving, a reliable and cutting-edge benchmark has foreseeable value in the field of deepfake forensics.

**Weaknesses:**

While the authors present up to 13 findings, these findings are not impressive enough. There are some shortcomings in the claimed findings including:
*   Some findings seem to simply reassert well-known motivations from previous works, such as Finding 1 with MAT(Zhao et al, CVPR 2021).
*   Some findings are common sense, such as Finding 2&7.

Furthermore, it is recommended that authors consolidate a Findings list in the conclusion part, as the presented Findings are dispersed throughout the 36 pages of the whole paper, making it less friendly for reading.

**Questions:**

I hope the authors can provide more constructive insights for designing deepfake detection methods according to the proposed benchmark, such as:

*   For model-centric methods, what impact do various architectures (e.g., CNN vs. Transformers) or modalities (e.g., frequency vs. noise) have on different kinds of deepfakes?
*   For data-centric methods, which data augmentations or training strategies are beneficial for deepfake detection?

---

> ### Author Response · Authors · 2024-11-23
> **Rebuttal by Authors**
>
> Thanks for your diligent review and comments. We appreciate your recognition of the value in addressing limitations of previous datasets. Below are responses to each of your comments. (The anonymous website referred to below is https://anonymous.4open.science/r/G6No-E3DF).
>
> ***W1: a)13 findings are not impressive enough；b) Finding 1 simply reasserts well-known motivations from MAT. Finding 2&7 are common sense; c) Recommend consolidating a Findings list in the conclusion.***
>
> a) Sorry for the confusion. DeepFaceGen fills the gap in the field by incorporating both novel and traditional forgery methods. This allowed us to uncover novel findings not possible with previous benchmarks. We have focused on using DeepFaceGen for further exploration in deepfake detection. Below are the additional experiments which will be included in the final version:
> - ***Capturing Temporal Forgery Artifacts in Diffusion-Generated Images***: Introducing temporal information during the denoising process of diffusion improves forgery detection performance.
> - ***Investigating the Impact of Identity Information on Face Forgery Detection Performance***: Models that extract information directly from the data are affected by ID information, whereas reconstruction-based methods are not significantly influenced by ID information in their detection performance.
> -  ***Cross-domain Detection Performance from Face to Non-face Scenarios***: (1) Directly applying face AIGC to non-face AIGC presents substantial challenges. (2) Face detection models can adapt to identify non-face features when trained on similar generative frameworks. (3) Models focus primarily on facial forgery artifacts, which hinders their ability to generalize effectively.
> - ***Exploring Symmetry Differences Between Real and Prompt-Guided Data***：Focusing on symmetry differences could help capture the physiological discrepancies between real and prompt-guided generated data, providing a new approach for image/video face forgery detection.
>
> Detailed results are in Sections 1-4 of the anonymous website.
>
> b) Sorry for the confusion. MAT focuses on texture feature analysis for task-oriented data (Appendix E, lines 1382-1384). In contrast, we expand the analysis to include the novel prompt-guided data, revealing that mid-frequency features are more effective in detecting such data. Additionally, our approach incorporates Multi-frequency Feature Analysis and Multi-feature Fusion, going beyond just texture features.
>
> To the best of our knowledge, there has been no prior study systematically comparing the quality of task-oriented and prompt-guided data before DeepFaceGen.  Building on the 20 evaluation models on DeepFaceGen, we further tested data from different generation methods and frameworks, leading to Findings 2 & 7.
>
> c) Thanks for your valuable advice. We will include a Findings list in the conclusion section of the final version to make it easier for readers to access our experimental results.
>
> ***Q1: Various architectures (e.g., CNN vs. Transformers) or modalities (e.g., frequency vs. noise) have on deepfakes***
>
> We are impressed by the depth of your insightful comments.The following is our experimental exploration of the proposed aspects.
>
> ***Various architectures***: The use of different model architectures is undoubtedly valuable for the diverse development of deepfake detection. In fact, we are currently exploring the performance differences of various model architectures. Through extensive comparisons of CNN-based and Transformer-based detection models, we found that ***in the video forgery detection domain, Transformer-based models generally outperform CNN-based models.***
>
> ***Various modalities***：Through detailed experiments on frequency-based and noise-based methods, we observed the following: 1) ***Frequency-based methods struggle to capture the forgery fingerprints of prompt-guided data***. 2) ***Both noise-based and frequency-based methods*** primarily detect fingerprints specific to particular forgery frameworks, but these fingerprints ***lack generalizability across different frameworks***.
>
> More experimental results are in Sections 5-6 of the anonymous website.
>
> ***Q2: Beneficial data augmentations or training strategies***
>
> We conducted detailed experimental analyses from the perspectives of ***Geometric Transformations, Color and Intensity Adjustments***, and ***Image Quality Transformations***, including methods such as Rotation, Isotropic Resize, Random Brightness and Contrast, FancyPCA, Hue Saturation Value (HSV) Adjustment, Image Compression, and Centrifugal, Additive, and Affine Transformations. The experimental results demonstrate that the following two methods significantly improve detection model performance:
> ***(1)Training the networks on local patches,cropped from the image with no resizing.(2)Centrifugal, additive, and affine transformations as data augmentation.***
>
> More experimental results can be found in Section 7 of the anonymous website.

---

### Author Response · Authors · 2024-11-23
**Rebuttal by Authors**

We sincerely thank all reviewers for their thoughtful feedback and recognition of our work's significance. We appreciate the acknowledgment of DeepFaceGen as a ***comprehensive and large-scale benchmark*** ("reliable and cutting-edge with diverse generation methods" – G6No, UKvR, herE), its ***contributions to advancing face forgery detection*** ("integration of task-oriented and prompt-guided approaches with insightful findings" – herE, 3GRu), and its ***utility for the community*** ("valuable resource for real-world applications" – UKvR, herE). We are also grateful for the praise regarding the ***clarity and organization of our paper*** ("easy-to-follow and well-structured" – G6No, herE). Your encouragement motivates us to further improve and expand our contributions.

We have endeavored to consider the feedback as comprehensively as possible, leading to a revision process that significantly honed the paper. We have addressed every point in our responses and are happy to follow up on any aspect during the discussion phase. Specifically, ***we have tackled stated weaknesses (W) and Questions (Q) with detailed answers***.

Finally, we would like to express our appreciation once again for the reviewers' constructive comments and careful reading, which undoubtedly lead to enhancing the quality of our work.

---

### Author Response · Authors · 2024-11-28
**Comment by Authors**

Dear Reviewers,

We sincerely appreciate your valuable feedback during the review process. Your insights will undoubtedly enhance the quality of our work. We have recently added additional visualization experiments on DeepFaceGen to further validate and explore our findings. We look forward to your continued feedback and guidance!

Best regards,

The Authors of "A Large-scale Universal Evaluation Benchmark for Face Forgery Detection".

---

### Meta-Review · Area_Chair_dFHz · 2024-12-22

**Metareview:**

The paper introduces DeepFaceGen, a comprehensive and large-scale evaluation benchmark designed to quantitatively assess the effectiveness and generalizability of face forgery detection techniques. The benchmark dataset consists of real and fake images/videos collected from various sources, generated using both task-oriented and prompt-guided methods. The reviewers acknowledge that this benchmark is novel compared to existing alternatives. However, their main concern lies with the novelty of the findings and the experimental evidences provided to support them.

Most of the findings presented in the paper, based on experiments using the new dataset, have already been reported in the literature. This raises the question of whether introducing a new dataset is necessary if it fails to reveal insights that are not already known. Although the authors addressed this concern in their response, reviewer 3GRu did not acknowledge their reply, and I also found it unsatisfactory.

Additionally, the authors referred to specific sections of an anonymous website to address some reviewers' questions and comments. Unfortunately, these sections were unavailable. As none of the reviewers responded to the authors’ rebuttal, I was unable to determine whether the authors provided satisfactory answers due to the missing sections on the anonymous website.

**Additional Comments On Reviewer Discussion:**

The reviewers' main concern lies with the novelty of the finding and the experimental evidences provided to support them.  Most of the findings presented in the paper, based on experiments using the new dataset, have already been reported in the literature. This raises the question of whether introducing a new dataset is necessary if it fails to reveal insights that are not already known. This is a very valid issue raised by the reviewer 3GRu. The authors responded to this comment, and 3GRu did not acknowledge their response. However, I found the authors' response unsatisfactory.

---

### Decision · Program_Chairs · 2025-01-22

Reject